# Structural insights into the atypical filament assembly of pyrin domain-containing IFI16

Archit Garg ID[1], Ewa Niedzialkowska[2], Jeffrey J Zhou ID[1], Jasper Moh[1], Edward H Egelman ID[2] & Jungsan Sohn ID[1,3,4✉]

## Abstract

**In response to various intracellular stress or damage-associated signals, inflammasomes can be activated and trigger a pyroptotic cell death process through the sequential assembly of structurally compatible and interacting filamentous oligomers involving the pyrin domains (PYD) of important inflammasome components. The PYD-containing interferon-inducible protein 16 (IFI16) has been suggested as a viral DNA sensor that can induce inflammasome formation, but it also has other inflammasome-independent functions, including interferon production. Here, the cryo-EM structure of the filament assembled by the PYD of human IFI16 reveals a helical architecture distinct from inflammasome PYD filaments. In silico interface energy calculations suggest that the helical architecture of the IFI16^PYD filament prevents interactions with inflammasome PYD filaments. Biochemical and cell biology experiments consistently demonstrate that IFI16 does not directly interact with inflammasome pyrin domains. Together, our results provide insights into the structural basis of the inflammasome-independent functions of IFI16, and also show that strict architectural compatibility requirements for interactions contribute to the signal transduction specificity in inflammasome signaling.**

**Keywords** Interferon-inducible Protein 16; Innate Immune System; Inflammasomes; Cryo-EM; Pyrin Domain
**Subject Categories** Autophagy & Cell Death; Immunology; Structural Biology

## Introduction

Assembling supramolecular structures has emerged as a major theme in intracellular signaling pathways (Gama et al, 2024; Harris and Lim, 2001; Jaqaman and Ditlev, 2021; Kagan et al, 2014; Kobe et al, 2025; Nanson et al, 2019; Schaefer et al, 2018; Wu, 2013). Often coined as supramolecular organizing centers (SMOCs) and/ or signaling by cooperative assembly formations (SCAFs), these membrane-less-organelle-like entities act as central hubs for numerous essential cellular processes such as metabolism, division, and even death (Gama et al, 2024; Harris and Lim, 2001; Jaqaman and Ditlev, 2021; Kagan et al, 2014; Kobe et al, 2025; Nanson et al, 2019; Schaefer et al, 2018; Wu, 2013). A key aspect of their signaling mechanism is to sequentially assemble architecturally congruent filamentous oligomers via rather small protein domains (~100 amino acids, a.a.). Such highly coordinated multimeric interactions then provide the sensitivity and stability necessary to trigger switch-like responses (Gama et al, 2024; Harris and Lim, 2001; Jaqaman and Ditlev, 2021; Kagan et al, 2014; Kobe et al, 2025; Nanson et al, 2019; Schaefer et al, 2018; Wu, 2013). Here, we report how architectural complementarity can dictate the signaling specificity of filaments assembled by pyrin domains (PYDs), which act as the signaling module in inflammasome pathways.

Inflammasomes are supramolecular signaling platforms that play vital roles in host innate defense against various intracellular maladies, which include organelle damage, tumor formation, and pathogen invasion (Barnett et al, 2023; Sharma and Kanneganti, 2021; Sharma and de Alba, 2021). A unique aspect of inflammasomes is the signal transduction via the sequential assembly of filamentous oligomers (Barnett et al, 2023; Hochheiser et al, 2022; Lu et al, 2014; Matyszewski et al, 2021; Xiao et al, 2023). That is, upon detecting molecular signatures arising from intracellular maladies (e.g., viral DNA and dysfunctional organelles), inflammasome receptors undergo oligomerization and assemble filaments using their pyrin domains (PYDs) (Barnett et al, 2023; Hochheiser et al, 2022; Lu et al, 2014; Matyszewski et al, 2021; Xiao et al, 2023). These helical filaments are architecturally congruent and provide a template for the polymerization of the PYD of ASC (ASC^PYD), which is the central inflammasome adapter (ASC: apoptosis-associated speck forming protein containing caspase-recruiting domain (CARD)) (Hochheiser et al, 2022; Lu et al, 2014; Matyszewski et al, 2021; Xiao et al, 2023). ASC is composed of a PYD and a CARD, and the ASC^PYD filament assembly results in its CARD filament formation, which then leads to the polymerization and activation of caspase-1 (Barnett et al, 2023; Lu et al, 2014). The activated protease then executes key host defense processes such as maturation of pro-inflammatory cytokines and pyroptotic cell death (Denes et al, 2012; He et al, 2015).

[1]Department of Biophysics and Biophysical Chemistry, Johns Hopkins University School of Medicine, Baltimore, MD 21205, USA. [2]Department of Biochemistry and Molecular Genetics, University of Virginia, Charlottesville, VA 22098, USA. [3]Division of Rheumatology, Johns Hopkins University School of Medicine, Baltimore, MD 21205, USA. [4]Department of Oncology, Johns Hopkins University School of Medicine, Baltimore, MD 21205, USA. ✉E-mail: jsohn@jhmi.edu

Both PYDs and CARDs are 6-helix bundles and belong to the death-fold (DF) superfamily (Park et al, 2007). Multiple cryo-EM studies revealed that all known PYD filaments share the same helical architecture (hexameric base, six protomers at the axial pole) (Hochheiser et al, 2022; Lu et al, 2014; Lu and Wu, 2015; Matyszewski et al, 2021; Morrone et al, 2015; Shen et al, 2019), while the CARD-subfamily filaments display their own common architecture (tetrameric base, four protomers at the axial pole) (Fu et al, 2016; Gong et al, 2021; Li et al, 2018; Lu et al, 2016; Matyszewski et al, 2018b; Wu et al, 2014; Zahid et al, 2020). Given that several upstream PYDs (and CARDs) can signal through one common downstream filament (Kagan et al, 2014; Lu and Wu, 2015; Morrone et al, 2015; Wu et al, 2014), such subfamily-specific congruent architectures provide an efficient framework for signal integration and transduction (Kagan et al, 2014). Moreover, recent studies have identified that the self-assembly of PYD and CARD filaments occurs bidirectionally, while the recognition between two signaling partners occurs unidirectionally, allowing for efficient activation and specific signal transduction (Hochheiser et al, 2022; Matyszewski et al, 2021). Nonetheless, the complexity of their interaction and assembly mechanisms is increasingly more appreciated, as DF proteins can utilize different interfaces to assemble and interact with their signaling partners (Gong et al, 2021; Robert Hollingsworth et al, 2021). Of note, it remains unknown whether there exist PYD- and/or CARD filaments whose structures deviate from their subfamily-specific architectures, and what such noncanonical assemblies would entail in dictating their signaling partners.

Interferon-inducible protein 16 (IFI16) is a member of absent-in-melanoma-2 (AIM2)-like receptors (ALRs), which contain a PYD and one or two nucleic acid-binding HIN200 domains (hematopoietic interferon-inducible nuclear antigen with 200 a.a.; Fig. 1A) (Barnett et al, 2023; Jin et al, 2012; Ni et al, 2016; Roberts et al, 2009). IFI16 acts as a versatile innate immune sensor that detects dysregulated nucleic acids (Unterholzner et al, 2010), with its most prominent mechanism of action being assembling filaments on unchromatinized double-stranded (ds)DNA (Antiochos et al, 2018; Garg et al, 2023; Morrone et al, 2014; Stratmann et al, 2015; Unterholzner et al, 2010). Upon assembly, IFI16 mediates a wide range of innate immune responses such as accentuating cGAS-STING-dependent interferon signaling pathways (Almine et al, 2017; Chang et al, 2024; Jonsson et al, 2017; Justice et al, 2021; Li et al, 2013; Li et al, 2012; Orzalli et al, 2015), viral replication restriction (e.g., herpes simplex viruses, cytomegalovirus, and human papilloma virus) (Hotter et al, 2019; Jakobsen and Paludan, 2014; Jiang et al, 2021b; Lo Cigno et al, 2015; Orzalli et al, 2013), and even chromatin dynamics regulation during DNA damage responses (Dunphy et al, 2018; Jakobsen and Paludan, 2014; Jiang et al, 2021a).

Currently, how and by what signaling partners IFI16 interacts within these various key innate immune functions remains largely speculative. For example, a long-standing conundrum about IFI16 is whether it directly activates ASC-dependent inflammasomes. Several studies independently showed that AIM2 is the only ALR that directly activates inflammasomes against cytosolic dsDNA by inducing the filament assembly of ASC (Fernandes-Alnemri et al, 2009; Gray et al, 2016; Hornung et al, 2009; Roberts et al, 2009). However, albeit lacking evidence for direct interactions, it was suggested that IFI16 may directly induce ASC-dependent

inflammasome formation against Kaposi's sarcoma herpes virus (KSHV) and human immunodeficiency virus (HIV) (Kerur et al, 2011; Monroe et al, 2014). Nevertheless, we recently found that recombinant full-length IFI16$^{(FL)}$ fails to induce the polymerization of ASC$^{PYD}$ regardless of bound nucleic acids (e.g., dsDNA, dsRNA, ssDNA, and ssRNA) (Garg et al, 2023), supporting the idea that this nucleic acid sensor is unlikely to interact with ASC$^{PYD}$ directly.

We present here a cryogenic electron microscopy (cryo-EM) structure of the IFI16$^{PYD}$ filament at 3.3 Å, which shows a helical architecture distinct from inflammasome PYD filaments (Hochheiser et al, 2022; Lu et al, 2014; Matyszewski et al, 2021; Shen et al, 2019). This atypical assembly is largely accomplished by the a.a. composition in two areas that alter the secondary structure of the IFI16$^{PYD}$ monomer. Interface energy calculations using *Rosetta* suggest that the helical architecture of the IFI16$^{PYD}$ filament is indeed incompatible with ASC$^{PYD}$ and its interacting partners. Moreover, in vitro and *in cellulo* experiments support our structure and in silico predictions that IFI16 interacts with neither ASC nor AIM2. Together, our results provide the structural basis for not only the versatile functions of IFI16 outside of inflammasomes, but also understanding how DF filaments regulate their signaling partners by altering helical architectures between and within subfamilies.

## Results

### The cryo-EM structure of the IFI16$^{PYD}$ filament

Although integral to understanding how IFI16 can operate outside of inflammasomes (Almine et al, 2017; Chang et al, 2024; Hotter et al, 2019; Jakobsen and Paludan, 2014; Jiang et al, 2021a; Jonsson et al, 2017), the structure of the IFI16$^{PYD}$ filament remains unknown. After extensive optimization, we identified conditions that allow recombinant IFI16$^{PYD}$ to auto-assemble into filaments (Fig. 1B). The average power spectrum from cryo-EM images of 256-pixel-long nonoverlapping filament segments showed that the IFI16$^{PYD}$ filament is a three-start helix with 134.6° rotation and 5.6 Å rise, displaying no rotational symmetry (Figs. 1C and EV1). Because the structure of IFI16$^{PYD}$ monomer is unknown, we used AlphaFold (Jumper et al, 2021) for model building. The resolution of the final model was 3.3 Å according to the gold standard method (Fig. 1D,E). The diameter of the outer rim is 85 Å and the inner cavity is 22 Å (vs. 94 Å and 25 Å in ASC$^{PYD}$) (Fig. 1F). Our structure shows that the assembly is mediated by the six interfaces that are commonly found in PYDs and CARDs (Types 1-3 ref. (Lu et al, 2014); Fig. 1Ga). The Type 1a:1b interface is composed of polar and hydrophobic residues (Fig. 1Gb), and Type 2a:2b and Type 3a:3b interfaces include mostly polar and ionic side-chain interactions as predicted previously (Fig. 1Gc,d) (Lum et al, 2019). In addition, we noted the side chains that were previously identified important for the filament assembly and dsDNA-mediated oligomerization of full-length IFI16 (e.g., Leu[28] and Ile[46]) (Liu et al, 2023; Lum et al, 2019; Morrone et al, 2014).

Notably, unlike the inflammasome filaments assembled by ASC$^{PYD}$ and its interacting partners, such as AIM2$^{PYD}$ (Lu et al, 2014; Matyszewski et al, 2021), the IFI16$^{PYD}$ filament lacks the C3 symmetry and displays vastly different helical parameters (Fig. 2A). Moreover, instead of the six protomers that constitute the

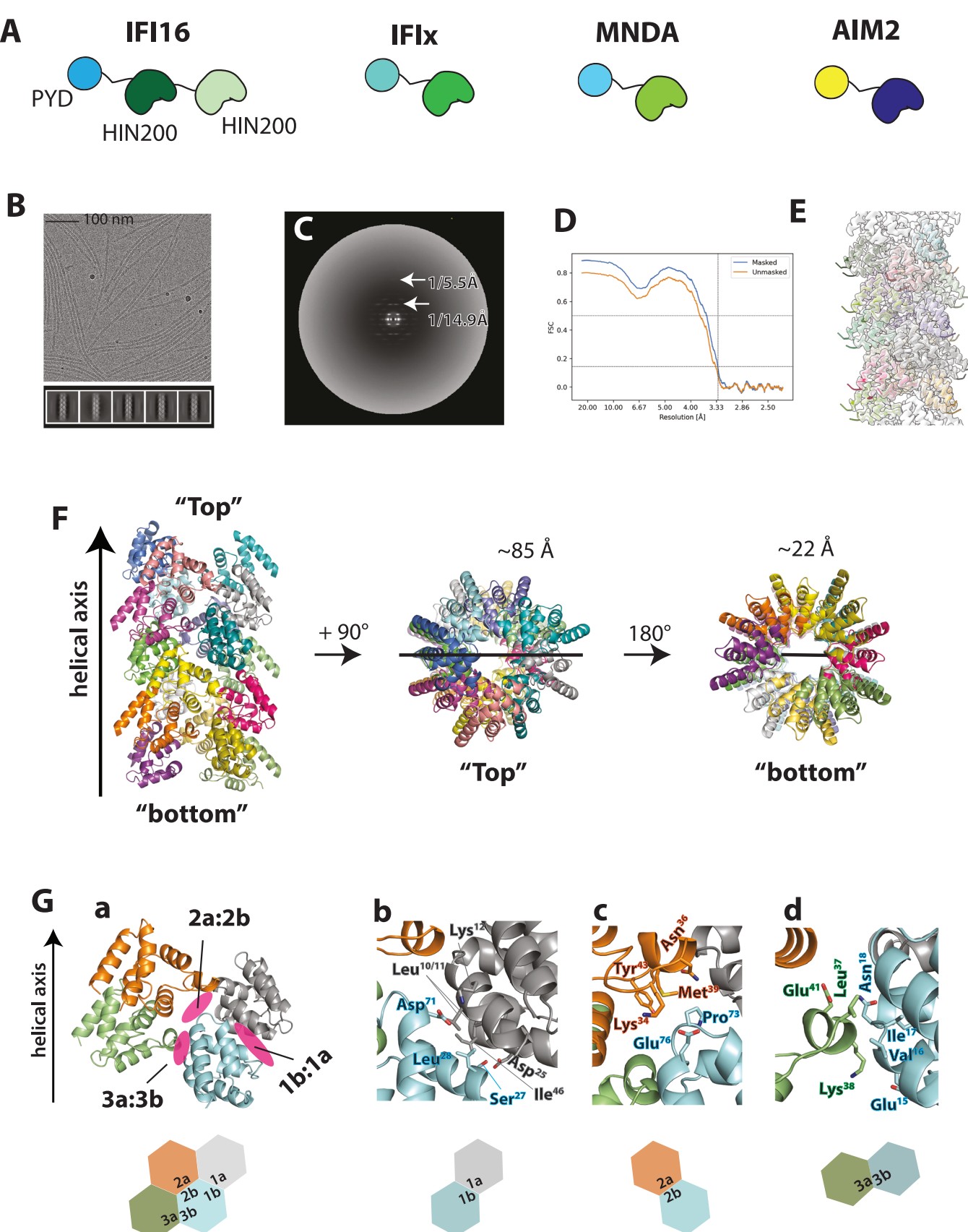

**Figure 1.   The cryo-EM structure of the IFI16$^{PYD}$ filament.**

(A) The ALR family members and their domain organization. Two or three functional domains (PYD and HIN200) are flanked by 50 to 100 a.a. linkers. MNDA: myeloid cell nuclear differentiation antigen. (B) A sample cryo-electron micrograph of the IFI16$^{PYD}$ filaments. 2D classes are shown below. (C) The average power spectrum of the IFI16$^{PYD}$ filament. (D) The FSC curve of the cryo-EM map of the IFI16$^{PYD}$. The dotted lines indicate the 0.143 threshold for the resolution. (E) The IFI16$^{PYD}$ filament model built into the cryo-EM map. (F) The atomic model of the IFI16$^{PYD}$ filament, with each subunit colored differently. (G) The three unique filament interfaces are shown along with the cartoon representation below. (b–d) Side chains that mediate protein-protein interactions at each unique interface are shown.

filament base of ASC$^{PYD}$ (Lu et al, 2014), the base of the IFI16$^{PYD}$ filament is composed of five protomers (i.e., hexagon vs. skewed pentagon; Fig. 2A,B). As a consequence, the diameters of both outer and inner rims of the IFI16$^{PYD}$ filament are smaller than those of the ASC$^{PYD}$ filament (Fig. 2B). In addition, although the top and bottom surfaces of both IFI16$^{PYD}$ and ASC$^{PYD}$ filaments are consistently electro-negative and positive, respectively, the charges are concentrated around the outer rim in the IFI16$^{PYD}$ filament, while they are either evenly distributed (negative) or follow the C3 symmetry (positive) in the ASC$^{PYD}$ filament (Fig. 2C).

To further compare the architecture of the IFI16$^{PYD}$ filament to the ASC$^{PYD}$ filament, we then generated the "honeycomb" side view of PYD filaments in which the middle subunit makes all repeated contacts with surrounding subunits for assembly (Fig. 2Db shows the corresponding honeycomb diagram with the six interfaces labeled) (Matyszewski et al, 2021; Wu et al, 2024). As expected, ASC$^{PYD}$ and AIM2$^{PYD}$ filaments align well with the monomers showing minimal deviations (RMSD: 0.8 Å; Fig. 2Da). By contrast, although the IFI16$^{PYD}$ monomer is still homologous to ASC$^{PYD}$ (RMSD: 1.5 Å), the filament organization deviates drastically when aligned at the center protomer (Fig. 2Dc).

Further examination indicated two key structural differences in the IFI16$^{PYD}$ monomer that would clash with ASC$^{PYD}$-like assembly. For instance, compared to ASC$^{PYD}$, the short helix 3 that mediates the Type 3 interface protrudes from the center of the protein by ~ 6 Å in IFI16$^{PYD}$ (dotted red circle in Fig. 2Dc). In addition, the loop between helices 5 and 6 in the Type 2 interface also extends from the core by ~ 5 Å due to Pro$^{73}$ that shortens the helix 5 and lengthens the helix 6 (dotted red circle in Fig. 2Dc). These differences in IFI16$^{PYD}$ would then clash with the ASC$^{PYD}$-like assembly in the Type 3 and Type 2 interfaces (Fig. 2Dd), also causing the Type 1 interface to misalign (Fig. 2Dc, red arrow).

## The atypical assembly of the IFI16$^{PYD}$ filament is encoded in the a.a. sequence

Comparing a.a. sequences of well-known ALRs revealed that the Pro$^{73}$ of IFI16 is conserved in all but AIM2 (Figs. 1A and 3A). On the other hand, the a.a. sequences of the helix 3 and connecting loops are poorly conserved in ALRs (Fig. 3A). Nonetheless, in contrast to the different conformations seen between IFI16 and ASC (and AIM2; Fig. 2D), the helix 3 of the IFI16$^{PYD}$ filament and that of the MNDA$^{PYD}$ monomer aligned well (RMSD = 0.7 Å; Figs. 1A and 3B). These observations indicate that the atypical assembly of the IFI16$^{PYD}$ filament is encoded in the a.a. composition of the monomer, leading to the changes in its secondary structure (filament assembly likely stabilizes the flexible helix 3).

Despite the poorly conserved a.a. sequences (Fig. 3A), we had previously found several common side chains between AIM2 and IFI16 that promote filament assembly (Morrone et al, 2014). While

inspecting our structure, we noticed several unique IFI16 side chains at filament interfaces that are conserved only in ALRs related more closely to IFI16, but not in AIM2 (Fig. 3A,C, highlighted in yellow, MNDA and IFIx). To further validate our structure and test the role of these subfamily-specific residues, we mutated them into AIM2 residues, then monitored the filament assembly of C-terminally mCherry-tagged IFI16$^{PYD}$ when transfected into HEK293T cells (Fig. 3D). Of note, we and others found that an N-terminal protein tag interferes with the filament assembly of DFs, while a C-terminal tag is inert (Lu et al, 2015; Matyszewski et al, 2018b; Matyszewski et al, 2021). Here, all mutations except M39K abrogated the filament assembly of IFI16$^{PYD}$-mCherry (Fig. 3D,E). Moreover, compared to single substitutions, simultaneously changing two interacting residues into those of AIM2's partially preserved the oligomerization activity of IFI16$^{PYD}$ (K12T/ D71K, Figs. 3C and EV2A). Together, these results not only support our cryo-EM structure, but also suggest that the a.a. composition of each ALR prescribes its distinct filament structure (see also (Lum et al, 2019)).

## IFI16 is incompatible with the inflammasome PYD-like assembly and vice versa

The structure of the IFI16$^{PYD}$ filament provides a compelling explanation as to why it would not directly interact with ASC. To further test the role of architectural complementarity, we then used *Rosetta* (Adolf-Bryfogle and Dunbrack, 2013; Chaudhury et al, 2010) to examine the energetics between IFI16$^{PYD}$ and other filaments involved in the inflammasome pathway (Matyszewski et al, 2021; Wu et al, 2024). Here, for self-assembly, the IFI16$^{PYD}$ honeycomb displayed favorable interface energy scores seen from other inflammasome PYD filaments (Matyszewski et al, 2021; Wu et al, 2024) (Fig. 4Aa,b). These scores were also markedly more favorable compared to when we modeled IFI16$^{PYD}$ into the AIM2$^{PYD}$ honeycomb in our previous study (Fig. 4Ab,c; see also ref. (Wu et al, 2024)). Conversely, when we modeled ASC$^{PYD}$ and AIM2$^{PYD}$ honeycombs using the IFI16$^{PYD}$ filament as a template, the resulting scores were dramatically less favorable compared to the scores from their native structures (Figs. 4B and EV2B; see also ref (Matyszewski et al, 2021; Wu et al, 2024)). Next, to evaluate how well ASC$^{PYD}$ can interact with the IFI16$^{PYD}$ filament compared to those assembled by its known upstream receptors, we replaced the center subunit of the receptor filament honeycombs with the central adapter (Fig. 4C). Here, NLRP3$^{PYD}$ and NLRP6 $^{PYD}$ honeycomb shells showed comparable energy scores toward ASC$^{PYD}$ with the "top" interface of these receptor assemblies preferred by the adapter as seen from the interaction with the AIM2$^{PYD}$ filament (Fig. 4Ca–c; see also ref. (Wu et al, 2024)). However, the interface energy scores resulting from ASC$^{PYD}$ in the IFI16$^{PYD}$ shell were markedly less favorable (Fig. 4Cd).

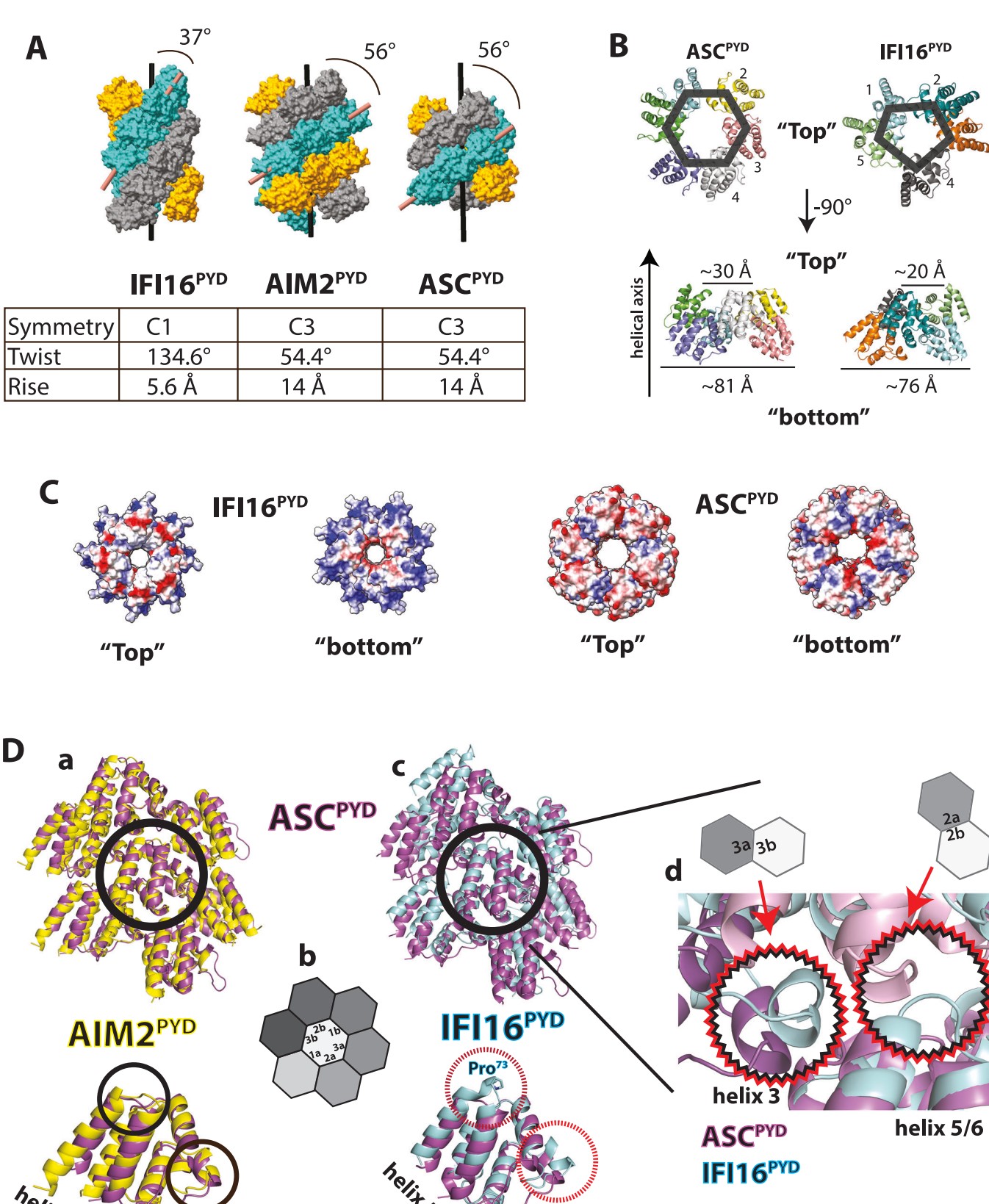

◀ **Figure 2. The architecture of the IFI16$^{PYD}$ filament is not compatible with the ASC$^{PYD}$ filament.**

(A) Comparison of the IFI16$^{PYD}$ filament to those assembled by AIM2$^{PYD}$ (PDB ID: 7k3r) and ASC$^{PYD}$ (PDB ID: 3j63). The angle between the central axis to the repeating helical strand is indicated. Key helical parameters are listed in the table. (B) Top- and side views comparing the ASC$^{PYD}$ and IFI16$^{PYD}$ filaments. A hexagon and a pentagon are shown to indicate the number of subunits at the filament base. (C) The electrostatic distribution of ASC$^{PYD}$ and IFI16$^{PYD}$ filaments. (D) (a) An overlay between AIM2$^{PYD}$ and ASC$^{PYD}$ filament "honeycombs" and monomers. The middle subunit is indicated with a black circle in the filament. The regions that AIM2$^{PYD}$ and ASC$^{PYD}$ do not deviate in the monomers are also indicated with black circles. (b) A schematic showing the "honeycomb" representation of PYD filaments. Each unique surface is noted in the center subunit, and the surrounding subunits are colored in different shades of gray. (c) An overlay of IFI16$^{PYD}$ and ASC$^{PYD}$ filament honeycombs and monomers. The middle subunit is indicated with a black circle in the honeycomb. The regions that deviate in the two proteins are indicated with dotted red circles in the monomers. (d) A zoom-in view of the two key areas that would clash between IFI16$^{PYD}$ and ASC$^{PYD}$ filament assemblies. The schematic diagrams in (c, d) show the location of clash/deviations using the interface identifiers shown in (b).

Consistent with our in silico analyses, when co-transfected in HEK293T cells as eGFP/mCherry-tagged proteins, all three inflammasome PYDs co-localized with ASC$^{PYD}$, while IFI16$^{PYD}$ failed to colocalize with the central adapter regardless of the amount of transfected plasmid (200–1200 ng; Figs. 4D and EV2C). We also co-transfected ASC$^{PYD}$ with two ALR PYDs that are closely related to IFI16 (Fig. 3A). Here, IFIx$^{PYD}$ formed filaments when transfected into HEK293T cells, while MNDA$^{PYD}$ remained diffused (Fig. EV2D). Nevertheless, consistent with the idea that IFI16-like ALRs do not directly interact with ASC, both proteins failed to colocalize with the ASC$^{PYD}$ filament (Fig. EV2D), Next, to test if the interaction between IFI16 and ASC is mediated by other domains beyond their PYDs, we tracked co-localization of full-length ASC$^{(FL)}$ and IFI16$^{PYD}$/IFI16$^{FL}$. Of note, when transfected into HEK293T cells, IFI16$^{FL}$ localizes in the nucleus (Wu et al, 2024), as seen from endogenous IFI16 from other cell types (Antiochos et al, 2018; Jiang et al, 2021a; Li et al, 2013; Li et al, 2012; Orzalli et al, 2013). Thus, we generated IFI16 variants to keep the full-length protein in the cytosol (ΔNLS: deletion of residues 96–100 and K128Q (Li et al, 2012)). Here, unlike AIM2$^{FL}$, ΔNLS- and K128Q-IFI16$^{FL}$ failed to colocalize with ASC$^{FL}$ (Fig. 4E; see also Fig. EV2E for different IFI16 plasmid amounts). In addition, the dsDNA-dependent oligomerization of recombinant IFI16$^{FL}$ and AIM2$^{FL}$ was also mutually exclusive as judged by the lack of FRET signals between donor-labeled AIM2 and acceptor-labeled IFI16 (Fig. EV3A). These results consistently suggest that IFI16 does not directly participate in the assembly of ASC-dependent inflammasomes.

Some helical filaments can adapt different symmetries during assembly while using the same oligomerization domains. For example, bacterial Pili filaments can assume the C5 symmetry or assemble without any rotational symmetry (Zheng et al, 2020). Moreover, while scaffolding toll/interleukin receptor (TIR) filaments assume parallel two-stranded open-ended assemblies, enzymatic TIR filaments assemble in an anti-parallel head-to-tail (Kobe et al, 2025; Nanson et al, 2019). A previous study showed that the AIM2$^{PYD}$ filament can also assemble without any rotational symmetry, albeit this architecture was caused by the N-terminal GFP-tag interfering with the native C3 symmetry (Lu et al, 2015). Of note, like the IFI16$^{PYD}$ filament, five subunits constitute the base of the GFP-AIM2$^{PYD}$ filament, but not six subunits in the untagged filament (Lu et al, 2015; Matyszewski et al, 2021). We thus compared the GFP-AIM2$^{PYD}$ filament structure to the IFI16$^{PYD}$ filament. Here, when we overlaid the two honeycombs, it was apparent that the overall assembly of the IFI16$^{PYD}$ filament is still different (Fig. EV3Ba). Moreover, the *Rosetta* energy scores resulting from modeling IFI16$^{PYD}$ using the GFP-AIM2$^{PYD}$

honeycomb were much less favorable than its native assembly (Fig. 4A vs. Fig. EV3Bb; around −40 vs. −13 *reus* for half-sum scores; see also ref. (Wu et al, 2024)). These observations consistently suggest that the architecture of the IFI16$^{PYD}$ filament is not accessible to inflammasome PYDs.

## Discussion

Inflammasomes play vital roles in host innate defense against various maladies ranging from pathogen invasion to the exposure to genotoxic stresses (Barnett et al, 2023; Sharma and Kanneganti, 2021; Sharma and de Alba, 2021). A key aspect of inflammasome pathways is the signal transduction by sequentially assembling homologs filaments (Barnett et al, 2023; Hochheiser et al, 2022; Lu et al, 2014; Matyszewski et al, 2021; Xiao et al, 2023). For instance, there are more than a dozen upstream receptors that recognize different types of molecular signatures associated with intracellular calamities (e.g., cytosolic dsDNA, malformed trans-Golgi network, and viral RNA) (Barnett et al, 2023). These initially divergent pathways then converge at the assembly of the central adapter ASC filament (Barnett et al, 2023). This is possible because, regardless of sensing mechanisms, all inflammasome receptors contain PYDs. Moreover, despite the a.a. sequence variance, all upstream PYDs assemble into architecturally congruent helical filaments to the ASC$^{PYD}$ filament, providing a structural template for assembly (Barnett et al, 2023; Hochheiser et al, 2022; Lu et al, 2014; Matyszewski et al, 2021; Xiao et al, 2023). We and others found that within these homologous suprastructures, there exist side-chain interactions that promote bidirectional self-assembly and the unidirectional recognition between upstream receptor PYD filaments and ASC$^{PYD}$ (Hochheiser et al, 2022; Matyszewski et al, 2021). Here, we determined the cryo-EM structure of the IFI16$^{PYD}$ filament, which revealed its unique helical assembly distinct from all other known PYD filaments. Our structure provides a mechanistic basis for its versatile innate immune functions outside of ASC-dependent inflammasomes (Almine et al, 2017; Antiochos et al, 2018; Chang et al, 2024; Hotter et al, 2019; Jakobsen and Paludan, 2014; Jonsson et al, 2017; Unterholzner et al, 2010) (Fig. 5).

There are four well-known ALRs: AIM2, IFI16, IFIx, and MNDA (Roberts et al, 2009; Unterholzner et al, 2010). The a.a. sequence of AIM2 (both PYD and HIN200) is most divergent from the other three ALRs (e.g., Fig. 3A), and IFI16 has an extra dsDNA-binding HIN domain (Roberts et al, 2009; Unterholzner et al, 2010). It is well-established that the primary function of AIM2 is to assemble the ASC-dependent inflammasome

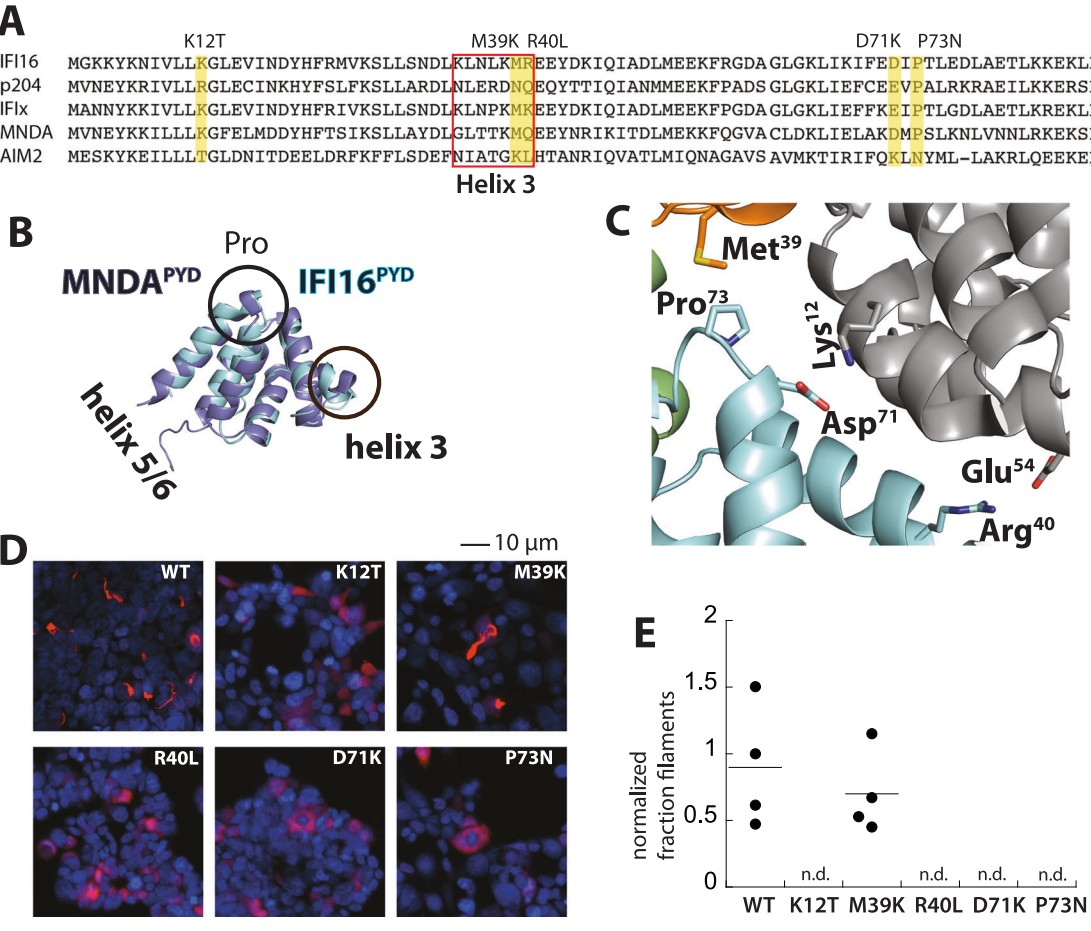

**Figure 3.** The atypical assembly of IFI16$^{PYD}$ is encoded in its a.a. sequence.

(A) The a.a. sequence alignment of ALRs (p204 = murine IFI16). (B) An overlay between MNDA$^{PYD}$ (PDB ID: 5h7q) and IFI16$^{PYD}$ monomers. (C) A cartoon showing the location of IFI16 residues we mutated into those in AIM2. (D) Images showing C-terminally mCherry-tagged IFI16$^{PYD}$ variants expressed in HEK293T cells (1200 ng). Blue: Hoechst. $n \geq 3$ biological replicates. (E) A plot showing the relative amount of filaments formed by the indicated IFI16$^{PYD}$ variants in HEK293T cells. $n \geq 3$ biological replicates. Source data are available online for this figure.

(Fernandes-Alnemri et al, 2009; Gray et al, 2016; Hornung et al, 2009; Matyszewski et al, 2018a; Matyszewski et al, 2021; Roberts et al, 2009). However, the other three ALRs represent a rare case for PYD-containing proteins whose biological functions are much more appreciated outside of inflammasomes (DeYoung et al, 1997; Mondini et al, 2007). For example, IFIX is a DNA sensor that has anti-viral and anti-tumor activities (DeYoung et al, 1997; Diner et al, 2015; Ding et al, 2004), and MNDA regulates the transcription of IRF7 by recruiting RNA polymerase II to its promoter (Gu et al, 2022). Most notably, IFI16 accentuates the cGAS-STING pathway, regulates viral latency and replication, suppresses oncogene expression, and controls chromatin dynamics upon DNA double-strand break (Almine et al, 2017; Antiochos et al, 2018; Chang et al, 2024; Hotter et al, 2019; Jakobsen and Paludan, 2014; Jonsson et al, 2017; Kerur et al, 2011; Unterholzner et al, 2010). Despite such versatile roles, how exactly IFI16 recognizes its signaling partners, and whether it can directly induce ASC-dependent inflammasome assembly remain controversial (Gray et al, 2016; Hornung et al, 2009; Kerur et al, 2011; Monroe et al, 2014; Roberts et al, 2009).

Combined with our previous observation in which recombinant IFI16$^{FL}$ fails to regulate the polymerization of ASC$^{PYD}$ regardless of bound nucleic acids (Garg et al, 2023), our results here further demonstrate that the IFI16 filament by itself does not directly participate in ASC-dependent inflammasome assembly. Instead, we speculate two possibilities for IFI16 to induce inflammasome formation via ASC. First, it is possible that there exist yet unidentified viral- or endogenous proteins that can bridge IFI16 and ASC (perhaps an ASC-like protein containing two PYDs that can assemble into different helical oligomers). Second, post-translational modification (PTM) of IFI16 triggered by different cues could alter the filament architecture, which might allow it to interact with ASC (four residues on IFI16$^{PYD}$ are subject to PTM: Lys$^{45}$ is known to be acetylated ((Li et al, 2012); Lys$^{6}$, Lys$^{34}$ and Lys$^{83}$ are predicted to be SUMOylated (PTMeXhange/Uniprot prediction; see also Fig. EV3C). Finally, considering that IFI16 and many other innate immune sensors undergo phase-separation (Du and Chen, 2018; Huoh and Hur, 2022; Liu et al, 2023; Liu et al, 2024; Shen et al, 2021; Wang and Zhou, 2024; Yu et al, 2021), it is possible that the role of IFI16 is to facilitate the condensation of

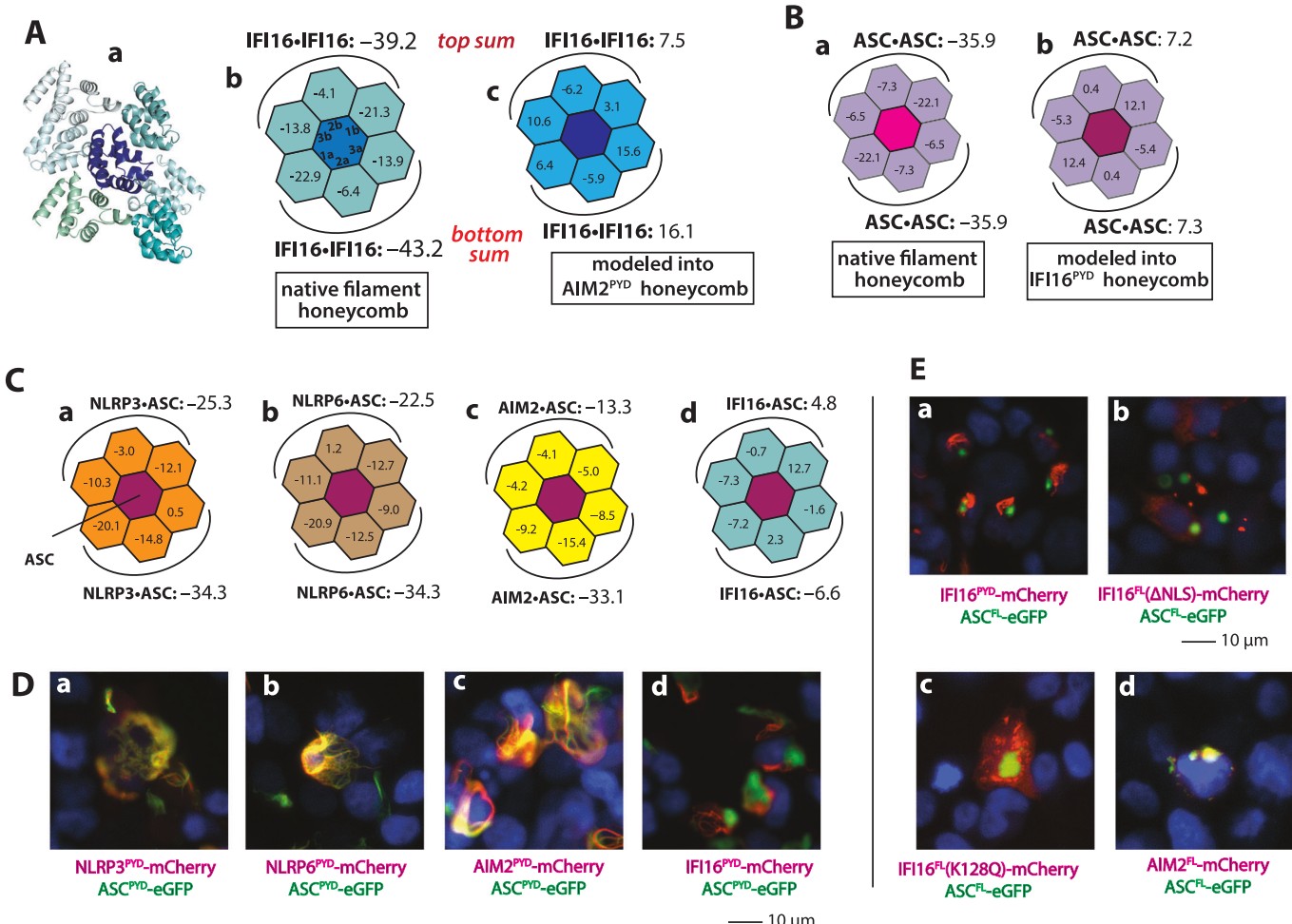

**Figure 4. IFI16^PYD does not directly interact with ASC^PYD and is incompatible with inflammasome PYD-like assembly.**

(A) (a) The honeycomb side view of the IFI16^PYD filament. The center protomer (dark blue) makes all six unique contacts with surrounding protomers for assembly. (b) The corresponding honeycomb diagram of IFI16^PYD filament with *Rosetta* interface energy units (*reus*). (c) The *reus* when the IFI16^PYD filament was modeled using the AIM2^PYD honeycomb (PDB: 7k3r). The sum of *reus* from the top and bottom halves are also shown. (B) The *reus* of the ASC^PYD honeycombs modeled when using (a) the native filament (PDB: 3j63) vs. (b) That of IFI16^PYD. (C) The resulting *reus* when ASC^PYD was modeled into the honeycomb shell of either its known upstream receptors (a–c) or (d) IFI16^PYD. (D) Fluorescence microscope images of HEK293T cells co-transfected with eGFP-tagged ASC^PYD (300 ng with inflammasome receptors and /600 ng with IFI16) and indicated mCherry-tagged plasmids (NLRP3^PYD:600 ng, AIM2^PYD, and NLRP6^PYD:300 ng, IFI16^PYD:1200 ng). Blue: Hoechst. *n* ≥3 biological replicates. (E) Fluorescence microscope images of HEK293T cells co-transfected with eGFP-tagged ASC^FL (600 ng) and indicated mCherry-tagged plasmids (IFI16^PYD:1200 ng, all others: 600 ng). Blue: Hoechst. *n* ≥3 biological replicates. Source data are available online for this figure.

inflammasomes and/or the cGAS-STING axis. Future studies directing these possibilities will elucidate how IFI16 can accomplish such versatile functions in regulating host innate immune responses.

PYDs belong to the DF superfamily and often found in proteins involved in inflammatory signaling pathways (Park et al, 2007). There are two well-known subclasses of PYDs. First, the vast majority of PYD-containing proteins are thought to act as the upstream receptor for ASC-dependent inflammasome formation (Barnett et al, 2023). The other class consists of pyrin-only-proteins (POPs) that interfere and regulate the assembly of various inflammasome PYD filaments (Devi et al, 2020; Wu et al, 2024). The structure of the IFI16^PYD filament presented here marks the third subclass of the PYD family, where it assembles into architecturally distinct filaments for their non-inflammasome

functions. Indeed, there is a subset of NLRPs whose role is to downregulate the activation of ASC-dependent inflammasomes (Chen et al, 2019; Cui et al, 2012). Determining the structures of these PYD filaments will further illuminate how the architectural diversity underpins the cellular functions of PYD filaments.

The limitation of our study is the use of the recombinant IFI16^PYD protein in isolation. Thus, it remains unclear how this filament assembly is mediated by the dsDNA-binding at the HIN200 domain. Prior studies have consistently demonstrated that the structural information obtained from isolated DF domains is highly biologically relevant (Gong et al, 2021; Hochheiser et al, 2022; Lu et al, 2014; Matyszewski et al, 2021; Shen et al, 2019; Xiao et al, 2023); however, future studies using the full-length protein bound dsDNA and/or in situ structural studies will further elucidate how IFI16 and other DF proteins operate in various cellular pathways.

# Methods

## Reagents and tools table

| Reagent/resource | Reference or source | Identifier or catalog number |
|---|---|---|
| **Experimental models** | | |
| DH5α competent cells (*E. coli*) | ThermoFisher | EC0112 |
| BL21 (DE3) cells (*E. coli*) | New England Biolabs | C2527I |
| HEK 293 T (*H. sapiens*) | ATCC | CRL-3216 |
| **Recombinant DNA** | | |
| pET21b IFI16$^{PYD}$-His$_6$ | Morrone et al, 2014 | N/A |
| pCMV6-IFI16$^{PYD}$ WT mCherry | Wu et al, 2024 | N/A |
| pCMV6-IFI16$^{PYD}$ K12T mCherry | This study | N/A |
| pCMV6-IFI16$^{PYD}$ M39K mCherry | This study | N/A |
| pCMV6-IFI16$^{PYD}$ R40L mCherry | This study | N/A |
| pCMV6-IFI16$^{PYD}$ D71K mCherry | This study | N/A |
| pCMV6-IFI16$^{PYD}$ P73N mCherry | This study | N/A |
| pCMV6-IFI16$^{PYD}$ K12T/D71K mCherry | This study | N/A |
| pCMV6-AIM2$^{PYD}$ mCherry | Wu et al, 2024 | N/A |
| pCMV6-ASC$^{PYD}$ eGFP | Wu et al, 2024 | N/A |
| pCMV6-NLRP3$^{PYD}$ mCherry | Wu et al, 2024 | N/A |
| pCMV6-NLRP6$^{PYD}$ mCherry | Wu et al, 2024 | N/A |
| pCMV6-IFI16$^{FL}$ K128Q mCherry | Li et al, 2012/ this study | N/A |
| pCMV6-IFI16$^{FL}$ ΔNLS mCherry | Li et al, 2012/ this study | N/A |
| pCMV6-ASC$^{FL}$ eGFP | Matyszewski et al, 2021 | N/A |
| pCMV6-AIM2$^{FL}$ mCherry | Matyszewski et al, 2021 | N/A |
| pCMV-SPORT6-PYHIN1 (IFIx) | Horizon discovery | MHS6278-202801470 |
| pCMV6-IFIx$^{PYD}$ mCherry | This study | N/A |
| pCMV6-MNDA$^{PYD}$ mCherry | This study | N/A |
| **Oligonucleotides and other sequence-based reagents** | | |
| PCR primers | This study | Dataset EV1 |
| **Chemicals, enzymes, and other reagents** | | |
| FD AgeI/BshTI | ThermoFisher | FD1464 |
| FD XhoI | ThermoFisher | FD0694 |
| T4 DNA ligase | New England Biolabs | M0202S |
| Phusion high-fidelity DNA polymerase | New England Biolabs | M0530S |
| FD DpnI | ThermoFisher | FD1704 |
| 100 mM dNTP | ThermoFisher | 10297018 |

| Reagent/resource | Reference or source | Identifier or catalog number |
|---|---|---|
| DNase I | ThermoFisher | EN0521 |
| Benzonase nuclease | Millipore Sigma | 70746 |
| DMEM high glucose, pyruvate | ThermoFisher | 11995065 |
| Fetal bovine serum (FBS) | ThermoFisher | 16140071 |
| Falcon Cell culture 12-well plate | Corning | 353043 |
| Lipofectamine 3000 | ThermoFisher | L3000008 |
| 1x PBS, pH 7.4 | ThermoFisher | 10010023 |
| NucBlue Live cell stain (Hoechst) | ThermoFisher | R37605 |
| Prolong Gold antifade mountant | ThermoFisher | P36930 |
| GeneJET Plasmid miniprep kit | ThermoFisher | K0503 |
| GeneJET Gel Extraction Kit | ThermoFisher | K0692 |
| DyLight 550 Maleimide | ThermoFisher | 62290 |
| DyLight 650 Maleimide | ThermoFisher | 62295 |
| Lacey carbon Cu grids | Electron Microscopy Sciences | LC400-Cu-25 |
| **Software** | | |
| BioTek Gen5 | Agilent | |
| Kaleidagraph | https://www.synergy.com | |
| SnapGene Viewer | https://www.snapgene.com/snapgene-viewer | |
| EPU | ThermoFisher | |
| CryoSparc v4 | https://cryosparc.com | |
| AlphaFold v2.3.2 | https://deepmind.google/science/alphafold/ | |
| UCSF Chimera | https://www.cgl.ucsf.edu/chimera/ | |
| Coot | https://www2.mrc-lmb.cam.ac.uk/personal/pemsley/coot/ | |
| Phenix | https://phenix-online.org | |
| Adobe Illustrator v29.5.1 | https://www.adobe.com/products/illustrator.html | |
| Rosetta v3.13 | https://rosettacommons.org | |
| **Other** | | |
| ÄKTA pure | Cytiva | |
| Vitrobot Mark IV | ThermoFisher | |
| Glacios TEM | ThermoFisher | |
| BioTek Cytation 5 | Agilent | |
| Tecan Infinite M1000 | Tecan | |

## Cloning, expression, and purification of recombinant IFI16$^{PYD}$

The human IFI16$^{PYD}$ construct (residues 1–94) was cloned into a pET21b vector with a C-terminal His$_6$-tag and transformed into *Escherichia coli* BL21$^{DE3}$ cells (see also (Morrone et al, 2014)). Protein expression was induced around OD$_{600}$: 0.5 using 0.2 mM isopropyl

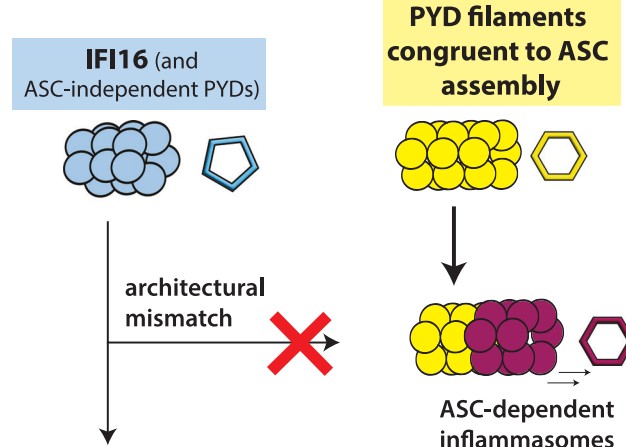

**Figure 5.** A cartoon describing how different PYD-containing proteins would and would not participate in ASC-dependent inflammasome assembly depending on their filament architectures.

β-D-1-thiogalactopyranoside at 18 °C for 16–18 h. The cells were resuspended homogenously and lysed by sonication in lysis buffer containing 25 mM NaPO$_4$, pH 7.8, 400 mM NaCl, 8 mM β-mercaptoethanol, 40 mM imidazole, and 10% glycerol. Cell lysate after centrifugation was loaded on a pre-equilibrated Ni$^{+2}$-NTA column and further purified using Superdex 75 16/600 size-exclusion column (Cytiva). Fractions containing IFI16$^{PYD}$ were pooled, concentrated to 95 μM, and stored in 20 mM HEPES pH 7.4, 400 mM NaCl, 1 mM EDTA, 1 mM DTT, and 5% glycerol at −80 °C.

## Cryo-EM sample preparation and data collection

The IFI16$^{PYD}$ filament sample was buffer exchanged to 40 mM HEPES pH 7.4, 160 mM NaCl, 1 mM EDTA, and 1 mM DTT and concentrated to 73 μM. In all, a 5 μl sample was applied to glow-discharged Lacey grids followed by auto-blotting for 4 s at 100% humidity and room temperature, which were then plunge frozen in liquid ethane using FEI Vitrobot Mark IV. Frozen grids were then clipped, and data were collected at the Beckman center for cryo-EM (Johns Hopkins School of Medicine) using the Thermo Scientific Glacios TEM operating at 200 kV and equipped with the Falcon 4i direct electron detector. In total, 3140 movies with a total dose of ~40 electrons/Å$^2$ were collected at a pixel size of 1.165 Å/pixel and a defocus range of −0.7 to −3.0 μm.

## Helical reconstruction and model building

The data were processed using CryoSPARC v4 (Punjani et al, 2017) (see also Fig. EV1). The 3140 movies, with a sampling of 1.165 Å/px, were subjected to "patch motion correction". The defocus values and astigmatism of the micrographs were determined by CTF estimation (patch CTF estimation) for the aligned full-dose micrographs. A total of 1072 micrographs were selected based on

relative ice thickness and CTF fit resolution for subsequent image processing. Filament tracer was used for segment picking from long filaments. The CTF-corrected micrographs were used for the segment extraction, with 256-pixel-long boxes, with a shift of 40 Å between adjacent boxes. These particles were subjected to 2D classification to identify classes containing single filaments and to remove bad segments. An averaged power spectrum was generated from rotationally aligned segments and used for determining the helical symmetry parameters. The selected particles were then processed with the helical refinement for the final reconstruction after the helical parameters (an azimuthal rotation of 134.6° and an axial rise of 5.6 Å per subunit) converged. The resolution of the final reconstruction was estimated by the FSC between two independent half maps, which showed 3.3 Å at FSC = 0.143.

We used AlphaFold v2.3.2 (Jumper et al, 2021) to generate a model based on the closely related MNDA$^{PYD}$ structure (PDB ID:5H7Q (Jin et al, 2017)) and docked into the IFI16$^{PYD}$ cryo-EM map by rigid body fitting, and then manually edited the model in Chimera (Pettersen et al, 2004) and Coot (Emsley et al, 2010). The IFI16$^{PYD}$ filament model was real-space refined using Phenix (Adams et al, 2010), and the final model was validated with MolProbity (Chen et al, 2010). Data collection parameters and refinement statistics are listed in Table 1.

## Cell culture and imaging

For cell imaging, each construct was cloned into the pCMV6 vector containing a C-terminal mCherry or eGFP tag. The mutations were introduced using a modified quick-change site-directed mutagenesis protocol (Liu and Naismith, 2008). Indicated plasmids were (co)-transfected into HEK293T cells at ~70% confluency (12-well plates) using Lipofectamine 3000 (Invitrogen). After 18 h, cells were washed with 1× PBS, fixed with 4% paraformaldehyde, followed by staining with Hoechst reagent (Invitrogen) and mounted on glass slides using ProLong Gold antifade reagent (Invitrogen). Imaging was performed using the Cytation 5 plate reader equipped with a fluorescent microscope (Agilent) and analyzed via Gen5 software (Agilent). The ratio of the amount of IFI16$^{PYD}$ filaments/cells was calculated for each construct and normalized to the WT. All used primers can be found in Dataset EV1. At least three biological replicates were conducted.

## Rosetta simulation

The InterfaceAnalyzer script in *Rosetta* was used to determine the interaction energy at individual interfaces of the honeycomb as described in (Adolf-Bryfogle and Dunbrack, 2013; Chaudhury et al, 2010; Matyszewski et al, 2021; Wu et al, 2024). We used the cryo-EM structures of IFI16$^{PYD}$ (PDB: 9muf), ASC$^{PYD}$ (PDB: 3j63), AIM2$^{PYD}$ (PDB: 7k3r), NLRP3$^{PYD}$ (PDB: 7pzd), NLRP6$^{PYD}$ (PDB: 6ncv), and GFP-AIM2 (PDB: 6mb2) filaments to generate corresponding honeycombs.

## Assays

At least three biological replicates of FRET-based quantitative assays using donor- and acceptor-labeled recombinant AIM2$^{FL}$ and IFI16$^{FL}$ were conducted as described previously (Matyszewski et al, 2018a; Mazanek and Sohn, 2019; Morrone et al, 2015; Stratmann et al, 2015).

**Table 1.  Cryo-EM data collection, model refinement, and validation statistics of the IFI16^PYD filament.**

| Data collection and processing | |
|---|---|
| Microscope | Glacios |
| Magnification | ×120,000 |
| Voltage (kV) | 200 |
| Electron exposure (e⁻ Å⁻²) | 40 |
| Defocus range (μm) | −0.7 to −3.0 |
| Pixel size (Å) | 1.165 |
| Helical rise (Å) | 5.6 |
| Helical twist (°) | 134.7 |
| Resolution (Å) at FSC threshold 0.143 | 3.3 |
| Map sharpening B factor (Å²) | −103.1 |
| **Refinement and model validation** | |
| Bond lengths rmsd (Å) | 0.002 |
| Bond angles rmsd (°) | 0.495 |
| MolProbity score | 1.74 |
| Clash score | 11.56 |
| Poor rotamers (%) | 0.00 |
| Ramachandran Favored (%) | 97.09 |
| Ramachandran Allowed (%) | 2.91 |
| Ramachandran Outliers (%) | 0.00 |
| **Data accessibility** | |
| EMDB accession code | EMD-48631 |
| PDB accession code | 9MUF |

## Data availability

The cryo-EM structure of the IFI16^PYD filament has been deposited to the protein data bank under accession code, PDB: 9MUF and the corresponding EM map was deposited in the EM Data Bank with accession code: EMD-48631. Image source data are included, and all primary data will be provided upon request.

The source data of this paper are collected in the following database record: biostudies:S-SCDT-10_1038-S44318-025-00626-7.

## Peer review information

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

## Acknowledgements

We thank the current and former Sohn lab members for discussion (especially Drs. Seamus Morrone and Mariusz Matyszewski). We also thank Dr. Ravi Sonani for data organization. This work was supported by NIH R35GM145363, NSF MCB1845003, NSF MCB2501395, and Johns Hopkins Accelerator Awards to JS; NIH R35GM122510 to EHE.

## Author contributions

**Archit Garg**: Conceptualization; Data curation; Formal analysis; Validation; Investigation; Visualization; Writing—original draft; Writing—review and editing. **Ewa Niedzialkowska**: Data curation; Formal analysis; Validation; Visualization; Methodology. **Jeffrey J Zhou**: Formal analysis; Investigation. **Jasper Moh**: Investigation. **Edward H Egelman**: Supervision; Funding acquisition; Methodology; Project administration. **Jungsan Sohn**: Conceptualization; Supervision; Funding acquisition; Writing—original draft; Writing—review and editing.

Source data underlying figure panels in this paper may have individual authorship assigned. Where available, figure panel/source data authorship is listed in the following database record: biostudies:S-SCDT-10_1038-S44318-025-00626-7.

## Disclosure and competing interests statement

The authors declare no competing interests.

# Expanded View Figures

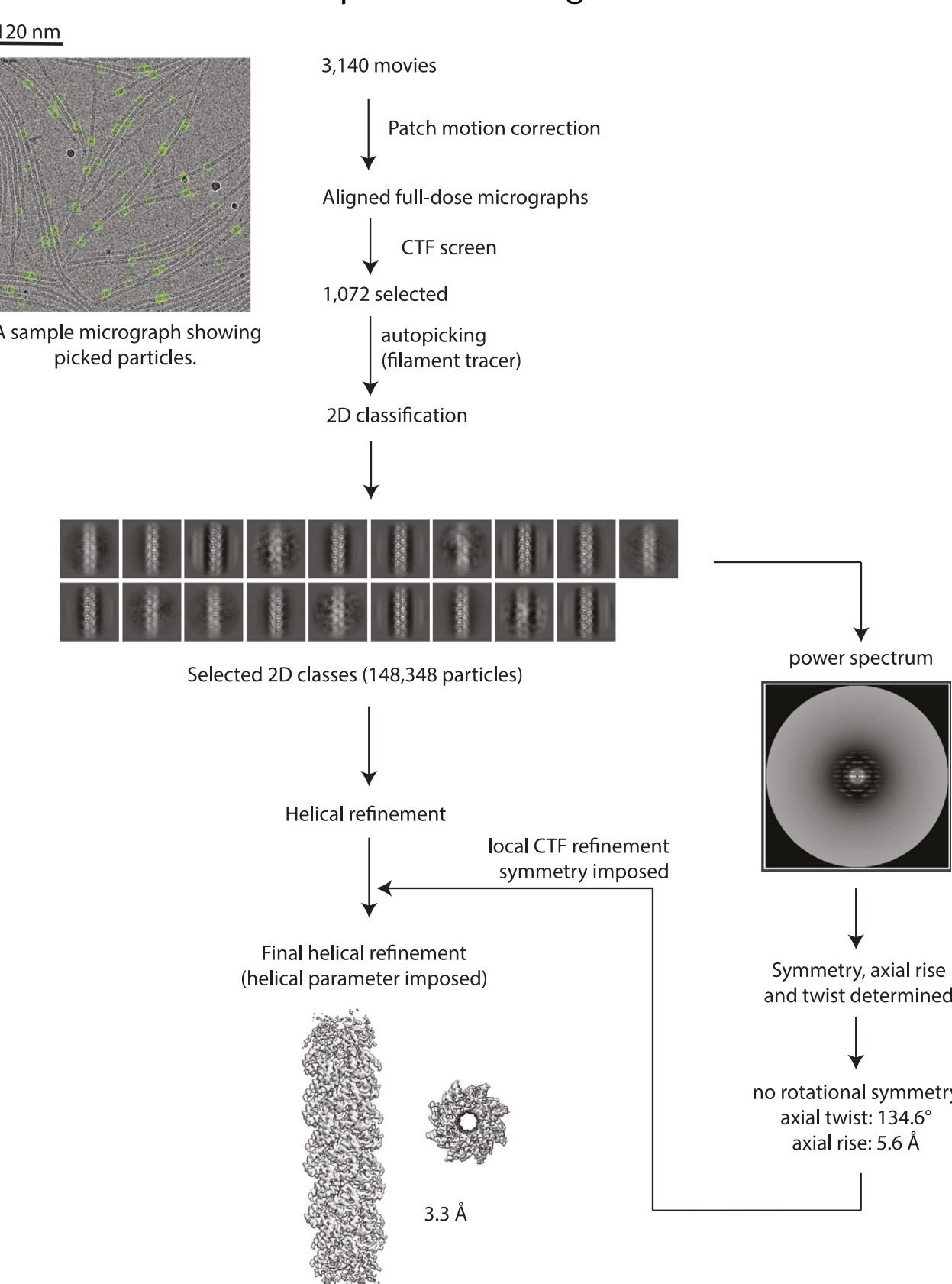

**Figure EV1.**   Cryo-EM image processing workflow for IFI16^PYD data detailing the processing from collected micrographs to the final map.

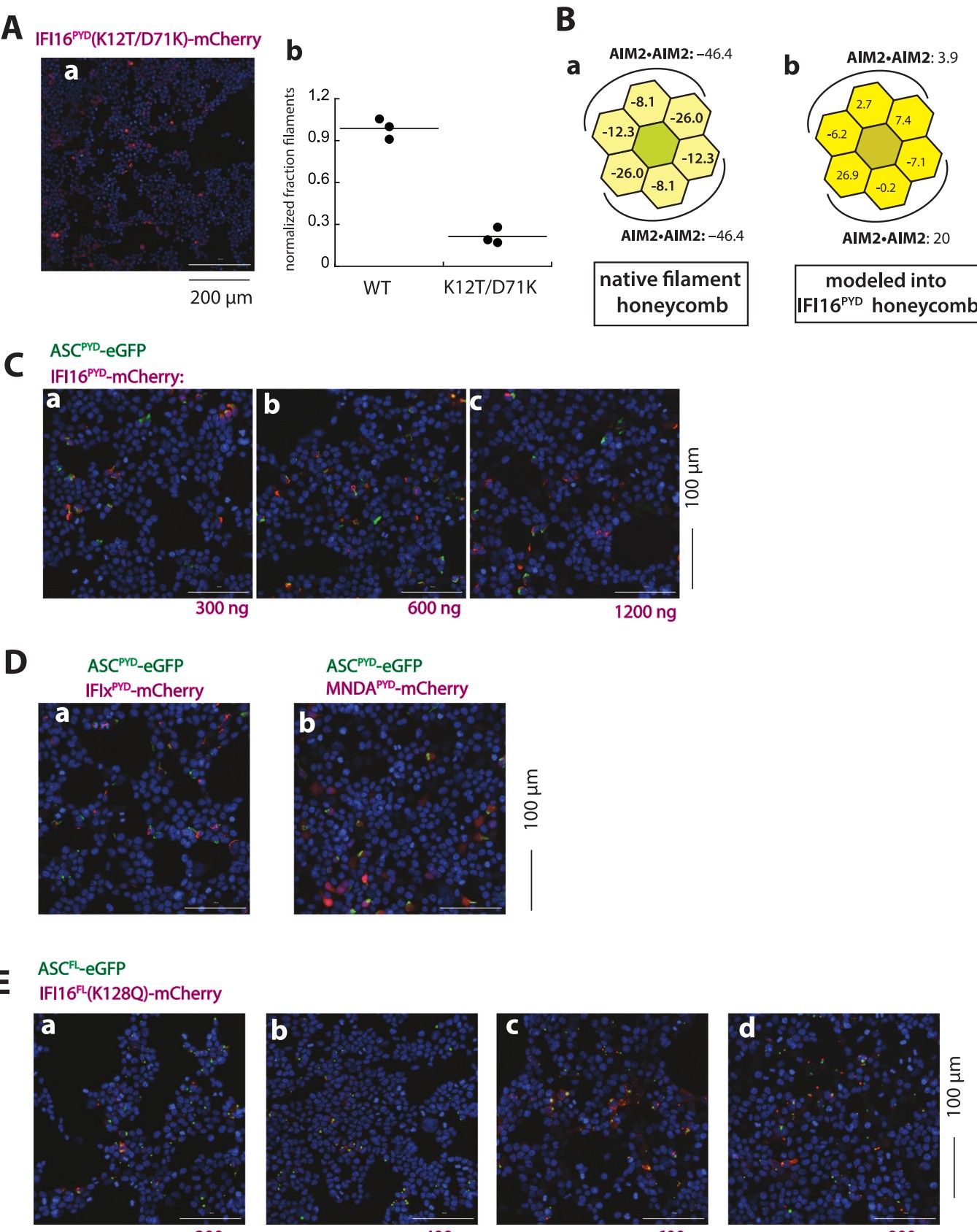

**A** IFI16^PYD(K12T/D71K)-mCherry

**b** normalized fraction filaments — WT, K12T/D71K

**B**

**a** AIM2•AIM2: −46.4 / −8.1 / −12.3 / −26.0 / −26.0 / −8.1 / −12.3 / AIM2•AIM2: −46.4 — native filament honeycomb

**b** AIM2•AIM2: 3.9 / 2.7 / 7.4 / −6.2 / −7.1 / 26.9 / −0.2 / AIM2•AIM2: 20 — modeled into IFI16^PYD honeycomb

**C** ASC^PYD-eGFP / IFI16^PYD-mCherry:

**a** 300 ng **b** 600 ng **c** 1200 ng — 100 μm

**D** ASC^PYD-eGFP / IFIx^PYD-mCherry — **a**

ASC^PYD-eGFP / MNDA^PYD-mCherry — **b** — 100 μm

**E** ASC^FL-eGFP / IFI16^FL(K128Q)-mCherry

**a** 200 ng **b** 400 ng **c** 600 ng **d** 900 ng — 100 μm

◀ **Figure EV2. Cellular and *Rosetta* analyses of IFI16^PYD filament assembly and its interaction with ASC.**

(A) (a) Fluorescence microscope image of HEK293T cells transfected with mCherry-tagged K12T/D71K-IFI16^PYD (1200 ng) Blue: Hoechst. (B) A plot showing the relative amount of filaments formed by indicated IFI16^PYD variants in HEK293T cells. $n \geq 3$ biological replicates. (b) The *reu*s of the AIM2^PYD honeycombs modeled when using (A) the native filament (PDB: 7k3r) vs. (B) that of IFI16^PYD. (C) Fluorescence microscope images of HEK293T cells co-transfected with eGFP-tagged ASC^PYD (600 ng) and the indicated amount of mCherry-tagged IFI16^PYD. Blue: Hoechst. $n \geq 3$ biological replicates. (D) Fluorescence microscope images of HEK293T cells co-transfected with eGFP-tagged ASC^PYD (600 ng) and mCherry-tagged IFIx^PYD and MNDA^PYD (both 600 ng) Blue: Hoechst. $n \geq 3$ biological replicates. (E) Fluorescence microscope images of HEK293T cells co-transfected with eGFP-tagged ASC^FL (600 ng) and the indicated amount of mCherry-tagged K128Q-IFI16^FL. Blue: Hoechst. $n \geq 3$ biological replicates.

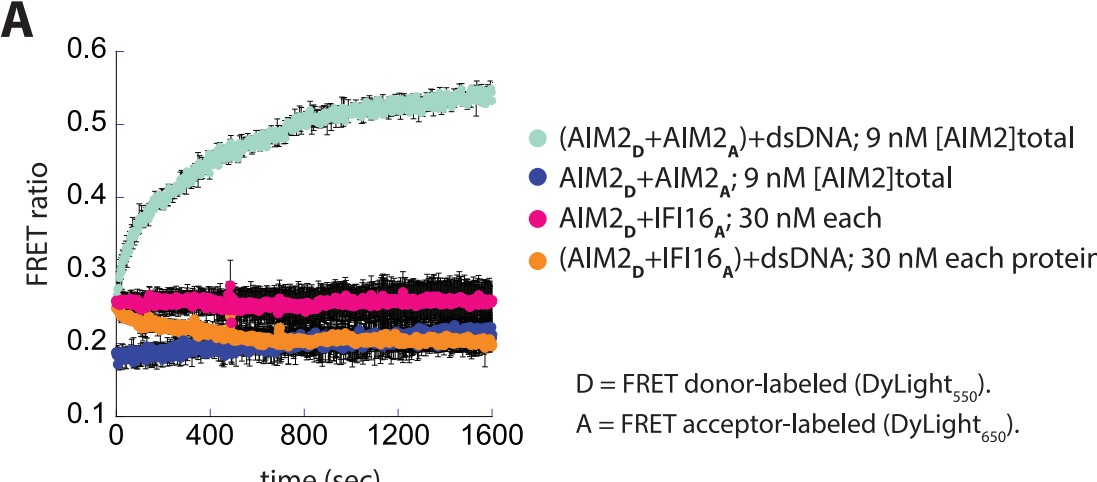

**A**

Legend:
- (AIM2$_D$+AIM2$_A$)+dsDNA; 9 nM [AIM2]total
- AIM2$_D$+AIM2$_A$; 9 nM [AIM2]total
- AIM2$_D$+IFI16$_A$; 30 nM each
- (AIM2$_D$+IFI16$_A$)+dsDNA; 30 nM each protein

D = FRET donor-labeled (DyLight$_{550}$).
A = FRET acceptor-labeled (DyLight$_{650}$).

**B**

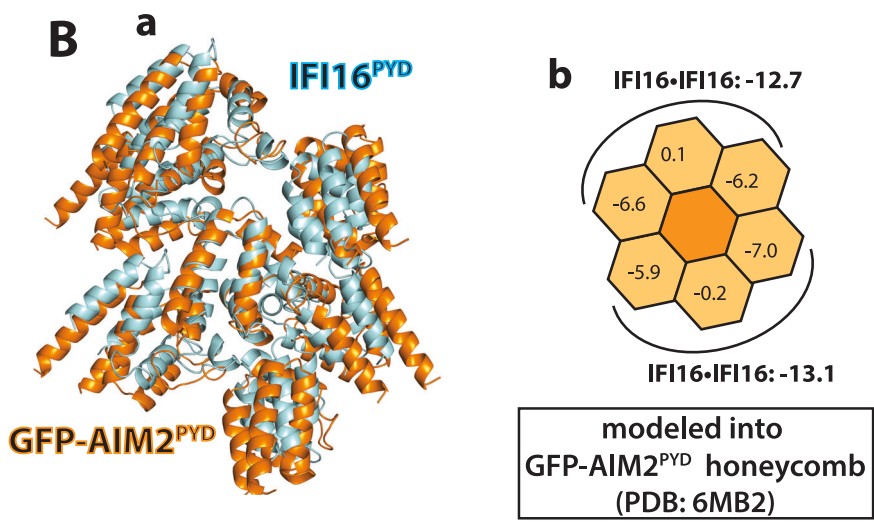

**a** IFI16$^{PYD}$ / GFP-AIM2$^{PYD}$

**b**
IFI16•IFI16: -12.7
0.1, -6.2, -6.6, -5.9, -7.0, -0.2
IFI16•IFI16: -13.1

modeled into
GFP-AIM2$^{PYD}$ honeycomb
(PDB: 6MB2)

**C**

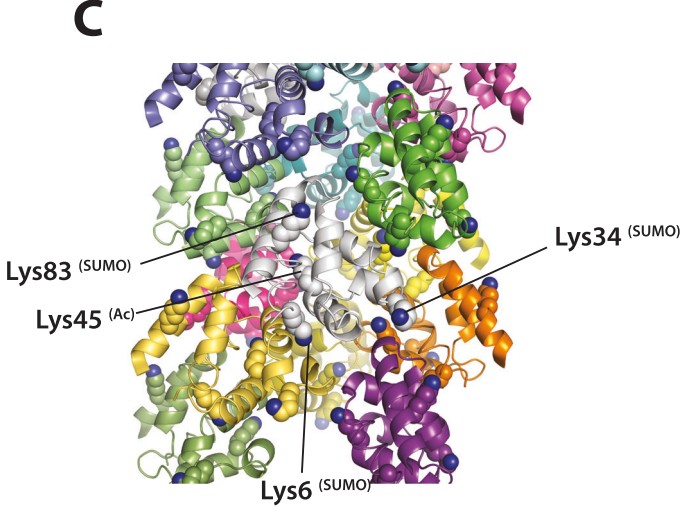

Lys83 (SUMO)
Lys45 (Ac)
Lys6 (SUMO)
Lys34 (SUMO)

◀  **Figure EV3.  IFI16 does not interact iwth AIM2.**

(A) A plot showing changes in the FRET ratio between indicated labeled proteins in the presence and absence of 600-bp dsDNA (10 µg/ml). Shown is the average of three independent experiments. Error bars = standard deviations. $n \geq 3$ biological replicates. (B) (a) An overlay between GFP-AIM2$^{PYD}$ (PDB ID: 6mb2) and IFI16$^{PYD}$ filaments. (B) The *reu*s of the IFI16$^{PYD}$ honeycomb when modeled using the GFP-AIM2$^{PYD}$ filament. (C) A cartoon showing the locations of putative (SUMOylation) and identified (acetylation) PTM sites on IFI16$^{PYD}$.

