## [Peer Review File · The EMBO Journal]

Structural insights into the atypical filament assembly of pyrin domain-containing IFI16

Archit Garg, Ewa Niedzialkowska, Jeffrey Zhou, Jasper Moh, Edward H. Egelman, and Jungsan Sohn

Corresponding author(s): Jungsan Sohn (jsohn@jhmi.edu)

Review Timeline:

Submission Date:	19th Feb 25
Editorial Decision:	25th Mar 25
Revision Received:	22nd Jun 25
Editorial Decision:	28th Aug 25
Revision Received:	9th Sep 25
Accepted:	28th Sep 25

Editor: Ioannis Papaioannou

Transaction Report:

Dear Dr. Sohn,

Thank you for submitting your manuscript EMBOJ-2025-120548 for consideration by The EMBO Journal, and for your patience during peer review. Your manuscript has now been seen by three experts in the field, and we have received the full set of their comments, which you can find below.

As you will see, the referees find the work interesting and well-performed, and the results novel and relevant, providing useful insights into the structural basis of IFI16 functionality. However, they also identify a number of limitations in the study and the manuscript.

In particular, they all raise concerns regarding the narrow focus of the text in the Introduction and Discussion, the insufficient background information provided for the non-specialist reader, the need to place the findings in a broader context, some overstatements that should be toned down, and numerous references that could or should be cited and discussed. We fully agree with the referees and would like to remind you that The EMBO Journal has a very broad readership of molecular biologists working in many different areas of life sciences, therefore articles published in our journal should be written in a way ensuring the highest possible level of accessibility and clarity not only to field specialists, but also to molecular biologists working in other fields. The significance and relevance of the findings should be made clear to our readership, without being overstated, while alternative interpretations and limitations of the work should also be sufficiently discussed, in line also with the very detailed and constructive referees' comments and suggestions.

Furthermore, there are concerns regarding the biological significance of the observed filament architecture (referee #2, point C), and the relevance of the biological context in the absence of any binding partners (referee #3, point 3), which are particularly relevant for our journal, and which would have to be addressed in a revision for the manuscript to be considered further for publication here. We also agree with referee #3 that the possibility of localization being affected by the relative overexpression levels (point 4) should also be addressed.

Given the referees' positive comments and recommendations, I would like to invite you to submit a thoroughly revised version of your manuscript taking the referees' recommendations on board, along with a detailed point-by-point response addressing all referees' comments. I should add that it is The EMBO Journal policy to allow only a single round of major revision, and acceptance of your manuscript will therefore depend on the completeness of your responses in this revised version. Please let me know if you have any questions or comments that you would like to discuss with me. If there are any major points you do not agree with or cannot address during your revision, I would encourage you to share them with me as early as possible to discuss how to proceed further in the most efficient way.

We generally allow three months as standard revision time (June 24, 2025). As a matter of policy, competing manuscripts published during this period will not negatively impact our assessment of the conceptual advance presented by your study. However, we request that you contact us as soon as possible upon publication of any related work, to discuss how to proceed. Should you foresee a problem in meeting this three-month deadline, please let us know in advance and we may be able to grant an extension.

Thank you for the opportunity to consider your work for publication in The EMBO Journal. I look forward to your revision.

Best regards,

Ioannis

Instructions for preparing your revised manuscript

1. When you are ready to submit the revision, please upload:

- A Word file of the manuscript text (including legends of main Figures, EV Figures and Tables). Please make sure that changes are highlighted (or "tracked") to be clearly visible.

- Individual production-quality figure files (one file per figure). When assembling your figures, please refer to our figure preparation guidelines in order to ensure proper formatting and readability in print as well as on screen:

If the data shown in a figure are obtained from n {less than or equal to} 2, please use scatter plots showing the individual data points.

i. the name of the statistical test used to generate error bars and P values

ii. the number (n) of independent experiments (please specify technical or biological replicates) underlying each data point (discussion of statistical methodology can be reported in the Materials and Methods section, but figure legends should contain a basic description of n , P , and the test applied)

iii. the nature of the bars and error bars (s.d., s.e.m.).

- A point-by-point response to the referees' comments, with a detailed description of the changes made (as a word file). All referees' concerns must be fully addressed and their suggestions taken on board. When preparing your letter of response to the referees' comments, please bear in mind that this will form part of the Review Process File and will therefore be available online to the community. Please note that you have the possibility to opt out of the transparent process at any stage prior to publication by letting the editorial office know (contact@embojournal.org); if you do opt out, the Review Process File link will point to the following statement: "No Review Process File is available with this article, as the authors have chosen not to make the review process public in this case.". For more details on our Transparent Editorial Process, please visit our website:

<https://www.embopress.org/page/journal/14602075/authorguide#transparentprocess>

- Expanded View (EV) files (replacing Supplementary Information) that are collapsible/expandable online. A maximum of 5 EV Figures can be typeset. EV Figures should be cited as "Figure EV1, Figure EV2" etc. in the text, and their respective legends should be included in the manuscript file after the legends of regular figures. See detailed instructions regarding Expanded View files here:

- For the figures that you do NOT wish to display as Expanded View figures, they should be bundled together with their legends in a single PDF file called "Appendix", which should start with a short Table of Contents (including page numbers). Appendix figures should be referred to in the main text as: "Appendix Figure S1, Appendix Figure S2" etc. Please see detailed instructions here: <https://www.embopress.org/page/journal/14602075/authorguide#expandedview>

- A complete author checklist, which you can download from our author guidelines

(<https://www.embopress.org/page/journal/14602075/authorguide>). Please note that the checklist will also be part of the Review Process File.

2. Please note that no statistics should be calculated and shown in Figures if $n=2$. Please also note that each p value should be reported as an exact value.

3. Before submitting your revision, primary datasets (and computer code, where appropriate) produced in this study need to be deposited in appropriate public databases (see <https://www.embopress.org/page/journal/14602075/authorguide#dataavailability>). The accession numbers, database, and the specific URLs (links) should be listed in a formal "Data availability" section (placed after Methods), following the example below:

"The RNA-seq datasets produced in this study are available in the following database:

Gene Expression Omnibus GSE46843 (<https://www.ncbi.nlm.nih.gov/geo/query/acc.cgi?acc=GSE46843>)"

*** All links should resolve to a page where the data can be accessed. ***

*** Please remember to provide in the Data availability section of your revised manuscript reviewer passwords if the datasets are not yet public. ***

*** The Data Availability Section is restricted to new primary data that are part of this study. In case you have no data that require deposition in a public database, please state so instead of referring to the database: "Our study includes no data deposited in public repositories." under the heading "Data availability". ***

4. The materials and methods need to be described in the manuscript using our structured methods format, which is now required for all research articles. According to this format, the Methods section includes a single "Reagents and Tools Table" - listing key reagents, experimental models, software and relevant equipment including their sources and relevant identifiers- followed by a "Methods and Protocols" section describing the methods. Please download and fill our Reagents and Tools Table template (.docx), which you can find in our author guide:

<https://www.embopress.org/page/journal/14602075/authorguide#structuredmethods>. When submitting your revised manuscript,

please do not include the Reagents and Tools Table in the Methods section of the manuscript but instead upload it as a separate file choosing the file type "Reagent Table".

5. Please check that the title and the abstract of the manuscript are brief, yet explicit, even to non-specialists. The length of the title should not exceed 100 characters, and the abstract should be a single paragraph not exceeding 175 words.

6. Please also note our reference format: <https://www.embopress.org/page/journal/14602075/authorguide#referencesformat>.

8. Please remember: digital image enhancement is acceptable practice, as long as it accurately represents the original data and conforms to community standards. If a figure has been subjected to significant electronic manipulation, this must be noted in the figure legend or in the "Materials and Methods" section. The editors reserve the right to request original versions of figures and the original images that were used to assemble the figure.

9. Our journal encourages inclusion of data citations in the reference list to directly cite datasets that were obtained from public databases. Data citations in the article text are distinct from normal bibliographical citations and should directly link to the database records from which the data can be accessed. In the main text, data citations are formatted as follows: "Data ref: Smith et al, 2001" or "Data ref: NCBI Sequence Read Archive PRJNA342805, 2017". In the Reference list, data citations must be labeled with "[DATASET]". A data reference must provide the database name, accession number/identifiers, and a resolvable link to the landing page from which the data can be accessed at the end of the reference. Further instructions are available at: <https://www.embopress.org/page/journal/14602075/authorguide#referencesformat>.

10. We request authors to consider both actual and perceived competing interests. Please review our policy (<https://www.embopress.org/page/journal/14602075/authorguide#conflictsofinterest>) and update your competing interests statement if necessary. Please name this section 'Disclosure and competing interests statement' and place it after the Acknowledgements section.

11. Please note that all corresponding authors are required to provide an ORCID ID upon submission of a revised manuscript (<https://orcid.org/>). Please find instructions on how to link your ORCID ID to your account in our manuscript tracking system in our Author guidelines (<https://www.embopress.org/page/journal/14602075/authorguide#authorshipguidelines>).

12. We use CRediT to specify the contributions of each author in the journal submission system. CRediT replaces the author contribution section, which should be removed from the manuscript. Please use the free text box to provide more detailed descriptions. See also guide to authors: <https://www.embopress.org/page/journal/14602075/authorguide#authorshipguidelines>.

14. We would also welcome the submission of cover suggestions or motifs to be used by our Graphics Illustrator in designing a cover.

15. Please use the link below to submit your revision:
<https://emboj.msubmit.net/cgi-bin/main.plex>

Referee #1:

Garg et al present interesting work on the pyrin domain (PYD) from the protein IFI16. The observation that it forms a filamentous structure different from other PYD and death-fold domains is significant, considering that so far death-fold domains seemed to form structurally compatible assemblies. The manuscript reads well and the work is technically sound.

The main comment I have is that the manuscript has a very narrow focus. Especially considering this journal having a broad readership in life sciences, it would be valuable to broaden the context of discussion and make the findings relate to a broader range of researchers. I suggest firstly that the Introduction includes a broader context of filamentous signalosomes and the implications for signalling (for example include references such as PMID 23582320 and 30517972,

doi.org/10.1042/BCJ20220094). The Discussion does briefly refer to signalling by filamentous assemblies (although not really to implications for signalling), but the references provided again have a rather narrow focus. In addition to expanding on the implications for signalling (refer to references mentioned above), it would be appropriate to mention the preprint characterizing all human death-fold domains (PMID 36993308), and discuss the implications of that work for signalling by the protein studied here. In the context of the main take-home message of the current work, it would further be useful to refer to the work on TIR domains that also form filamentous signalling assemblies, and feature different types of filamentous structures (congruent in some aspects but non-congruent in others; see PMID 34868065 and 39190506 for reviews). In fact, line 324-325 mentions bacterial pili filaments, which are not signalling assemblies and therefore much less relevant in some ways to the filaments studied here than TIR domain filaments.

Other specific comments are listed below.

1. Line 22-23: this is the first sentence of the abstract and although there is nothing inaccurate about what it says, I believe it will be difficult for most readers to unpack what it means, it would be nice to simplify it a bit.
2. Line 27: why does PYD have brackets? Please make this notation consistent throughout (e.g. line 102 also).
3. Line 27: the sentence should start with a capital letter.
4. Line 27-30: I believe this will also be rather unclear to most readers, several things from biology to methodology need more introduction so they can be more easily understood.
5. Line 63: considering the death domain is a specific type of domain in addition to being a member of the death-domain superfamily, I suggest distinguishing these two things by using a different abbreviation for the superfamily - e.g. "death fold (DF)"?
6. Line 128: which version of AlphaFold - they are very different from each other so be specific and use the most appropriate reference.
7. Line 131: "is at" doesn't sound right, please rewrite.
8. Line 148: "Side"
9. Line 171: "are" not "were".
10. Line 187: "Comparison of"?
11. Line 194: "where" rather than "that"?
12. Line 196: "schematic diagram".
13. Line 221 and throughout: despite widespread use, there is no such thing as "primary sequence" - there is no secondary sequence is there? Please do not be among the 100s of researchers who lack attention to detail and mix up "primary structure" and "amino-acid sequence" into a hybrid term that makes no sense. Just use "amino-acid" or "protein" sequence instead, is there anything wrong with those?
14. Line 259-278: what sort of predictions does one get when using AlphaFold Multimer to predict an oligomer of at least 6 monomers? This would be another way to analyze the differences between different PYDs, as long as the control predictions reproduce the experimental structures.
15. Line 385-390: is there any good evidence that IFI16 triggers ASC filament assembly at all (i.e. the third possibility would be it triggers the inflammasome using a different set of proteins)? I guess the first possibility with a "bridging" protein also involves an indirect link (although "bridging" paints a picture of a direct bridge). Please rewrite this discussion to allow all possibilities and refer to evidence why some may be more likely than others.
16. Line 468: I suggest "assays".
17. Several references have incomplete information: Antiochos, Devi, Lu, Wu S.
18. Supp Fig 1: the diagram makes no sense, the two parallel paths meet only at the structure - surely they don't lead to the same structure independently. Also make capitalization consistent.
19. Supp Fig 2: what is "Lorem ipsum"?

Referee #2:

The novelty of the study is best summarized in its discussion (line 365-368), that IFI16 together with two other members of the Aim2-like receptors (IFX and MNDA), are structurally and functionally distinct from Aim2, despite the amino acid sequence similarity.

In order to demonstrate this, the authors (1) engineered and reconstituted the PYD domain filamentous oligomer of IFI16, and use this platform to investigate the biological roles of IFI16; (2) resolved the recombinant filamentous oligomeric structure of IFI16 PYD, using helical reconstruction of cryo-EM data; (3) conducted surface and interface analysis of the filamentous structure of IFI16 PYD and other inflammasome PYD, including Aim2 PYD filamentous structures, and concluded that they are distinct and not compatible; (4) performed supporting biochemical and cellular analysis, including mutating key residues, to support conclusions in (3).

Considering the senior authors track record and expertise in helical reconstruction, as well as the rigorous structural information elaborated in this manuscript, the main conclusions (2) and (3) are convincing. This new structural insight revealed new alternative signaling complex formation and opened doors for further functional investigations in IFI16, IFX and MNDA. In

addition, this result would inspire more researchers to revisit the canonical linear signaling pathways that involves Death Domains in general.

Overall, several points are to be addressed/ to facilitate general audience to better appreciate this finding:

A. The authors were apparently very excited about this finding, reiterated some of the messages multiple times. Perhaps they would consider move line 93-101 in front of 82, and merge discussion line 340-360 together with the introduction, so that the key message could be emphasized directly in the discussion, since IFI16 PYD is concluded not relevant to ASC activation based on the provided data.

B. Divergent and convergent, as well as distinct surface pairing/seeding were seen in many CARD domain mediated signaling events, thus perhaps the statements in line 76-78 could be tuned down a bit to better reflect the current DD research landscape. I will recommend incorporating citations (Nature Communications volume 12, Article number: 189 (2021), figure 6; and Nature Communications volume 12, Article number: 188 (2021), figure 4). The authors also mentioned that the new interfaces might indicate the presence of novel interaction partners, perhaps they could shed light on such possibilities. (STING signaling pathway?)

C. Since most of the supporting evidence were lack of interaction with ACS inflammasome PYDs, negative data could also be explained by misfolding and non-ideal recombinant constructs. Filament formation of small globular domains sometimes could be completely driven by concentration or certain buffer conditions, like many enzymes and viral proteins. The authors are at the best positions to comment on potential oligomeric filamentous structures of IFI16 and MNDP PYD domains, any preliminary biochemical information could significantly strengthen the conclusion. Of course, complete dissection the biological significance of the observed new filament architecture is a highly demanding job, could be outside this manuscript's scope.

Referee #3:

Comments:

In this study, the authors utilize cryo-EM to characterize the filament structure of IFI16PYD and demonstrate its incompatibility with those assembled by the central inflammasome adaptor ASCPYD. Their model suggests that the atypical architecture of IFI16PYD filament makes it unlikely to directly interact with ASC, which is crucial for its noninflammasome functions. Overall, these findings provide insight into the structural basis of IFI16 multifunctionality. The study also importantly adds observations and comparisons that to help address existing inconsistencies in the literature about the extent to which IFI16 (in)directly participates in the assembly of ASC-dependent inflammasomes. However, this reviewer has the following concerns.

Major concerns:

1. The introduction needs to more comprehensively cover IFI16 structure and biology (paragraph on lines 82-104). The introduction would benefit from descriptions of IFI16 PYD features and properties. Citations should be included for more of the studies/groups that made initial discoveries of IFI16 "assembling filaments on unchromatinized dsDNA" (lines 82-84) and "accentuating cGAS-STING-dependent interferon signaling pathways, viral replication restriction, and even chromatin dynamics regulation during DNA damage responses" (lines 87-88). For example, more structural studies should be referenced on IFI16 binding to DNA through its HIN200 domains (ex. Jin et al., 2012, PMID: 22483801; Ni, et al., 2016, PMID: 26246511). cGAS-dependent stabilization of IFI16 was shown to facilitate innate immune responses during herpes simplex virus infection (Orzalli et al., 2015, PNAS). IFI16 was shown to be an antiviral innate immune factor and viral restriction factor, both earlier and in response to a much wider range of viruses than the included Almine, et al., 2017, Hotter, et al., 2019 and Jonsson, et al., 2017 papers (ex. Li, et al., 2012, PMID: 22691496; Orzalli, et al., 2012, PMID: 23027953 ; Li, et al., 2013, PMID: 24237704 ; Thompson, et al., 2014, PMID: 33986530 ; Cigno, et al., 2015, PMID: 25972554 ; Jiang, et al., 2021, PMID: 33986530). Primary literature on IFI16 association with DNA damage responses should be included (ex. Dunphy, et al., 2018, PMID: 3019309 ; Justice, et al., 2021, PMID: 34144993).

2. In their previous study (Garg et al., 2023), they show that IFI16FL is unable to promote the polymerization of ASCPYD regardless of the bound nucleic acids. In this study, is there a specific reason why they choose a condition where IFI16PYD, but not IFI16FL, auto-assembles into filaments without any additional molecules (e.g., nucleic acids) for the subsequent structural analysis?

3. Another concern is whether this represents a relevant biological context. Regardless of whether IFI16 functions to activate the inflammasome, in most cases, there are stimuli or binding partners that IFI16 interacts with to induce the downstream signaling. Aside from characterizing the IFI16PYD self-assembly filaments, do the authors think the IFI16PYD filaments would be similar in a cellular context with full-length IFI16 and binding molecules? Does IFI16 typically form distinct filaments in cells without any binding partners? In Figure 5, the authors propose a model illustrating how the atypical structure of IFI16 filaments may participate in non-inflammasome functions, including controlling gene expression, regulating chromatin dynamics, and restricting viral replication. These functions all require an interacting partner (primarily nucleic acids) with IFI16FL.

4. Figure 4D, 4E: The authors transfect different forms of ASC and IFI16 constructs into HEK293T cells and observe no colocalization. However, the localization state of the transgenes may be altered by their relative overexpression levels. It would be helpful to repeat the same experiment in cells with endogenous levels of IFI16 and ASC localized within the same cellular compartment. Otherwise, it would be helpful to perform titration experiments that could conceivably cover an expected range of the endogenous expression levels.

Minor points:

1. Based on their structural analysis of IFI16PYD filaments and existing literature on IFI16 post-translational modification, the authors should briefly comment on which modifications could potentially disrupt or reshape the helical architecture of the filaments, and which sites might be the most significant.

2. I believe the paper cited in line 285 (Li et al, 2013, Cell Host Microbe) should be a different paper, Li, et al., 2012, PNAS (PMID: 22691496).

3. Based on point mutations made in Figure 3, the authors determined that IFI16PYD filament assembly relies on its primary sequence. This is consistent with previous literature examining IFI16 PYD-dependent filamentation upon generating PYD point mutations (ex. PMID: 31337724 ; 37283074), which should be referenced. While the authors mutated side chains at the filament interfaces conserved in IFI16-related ALRs but not in AIM2, surrounding regions of the primary sequences are also not always similar between IFI16 and AIM2. It would be interesting to discuss or test whether replacement of partial/entire helices would restore IFI16PYD filaments (ex. expression of an IFI16 mutant with dual-replaced K12 and D71 helical regions).

We thank all three reviewers for their insightful comments. Please see below for our response to each point. We highlighted our edits in the revised manuscript with the following colors: per suggestion by **Reviewer 1**, **Reviewer 2**, and **Reviewer 3**.

Reviewer 1:

The main comment I have is that the manuscript has a very narrow focus. Especially considering this journal having a broad readership in life sciences, it would be valuable to broaden the context of discussion and make the findings relate to a broader range of researchers. I suggest firstly that the Introduction includes a broader context of filamentous signalosomes and the implications for signalling (for example include references such as PMID 23582320 and 30517972, doi.org/10.1042/BCJ20220094).

We added a new introductory paragraph. We added suggested citations. (lines 45-59)

The Discussion does briefly refer to signalling by filamentous assemblies (although not really to implications for signalling), but the references provided again have a rather narrow focus. In addition to expanding on the implications for signalling (refer to references mentioned above), it would be appropriate to mention the preprint characterizing all human death-fold domains (PMID 36993308), and discuss the implications of that work for signalling by the protein studied here. In the context of the main take-home message of the current work, it would further be useful to refer to the work on TIR domains that also form filamentous signalling assemblies, and feature different types of filamentous structures (congruent in some aspects but non-congruent in others; see PMID 34868065 and 39190506 for reviews). In fact, line 324-325 mentions bacterial pili filaments, which are not signalling assemblies and therefore much less relevant in some ways to the filaments studied here than TIR domain filaments.

Our apologies for failing to discuss such an important example. We added suggested points about TIRs along with the bacterial pili and added citations (lines 259-264).

Other specific comments are listed below.

1. Line 22-23: this is the first sentence of the abstract and although there is nothing inaccurate about what it says, I believe it will be difficult for most readers to unpack what it means, it would be nice to simplify it a bit.

We simplified this sentence in Abstract.

2. Line 27: why does PYD have brackets? Please make this notation consistent throughout (e.g. line 102 also).

This has been our convention to note that PYD is an acronym from the preceding part of the sentences (e.g., PMID: 33980849).

3. Line 27: the sentence should start with a capital letter.

Fixed.

4. Line 27-30: I believe this will also be rather unclear to most readers, several things from biology to methodology need more introduction so they can be more easily understood.

We simplified this sentence in Abstract.

5. Line 63: considering the death domain is a specific type of domain in addition to being a member of the death-domain superfamily, I suggest distinguishing these two things by using a different abbreviation for the superfamily - e.g. "death fold (DF)"?

We changed DD to DF.

6. Line 128: which version of AlphaFold - they are very different from each other so be specific and use the most appropriate reference.

We added the reference (line 409; v2.3.2).

7. Line 131: "is at" doesn't sound right, please rewrite.

Fixed.

8. Line 148: "Side"

Fixed.

9. Line 171: "are" not "were".

Fixed.

10. Line 187: "Comparison of"?

Fixed.

11. Line 194: "where" rather than "that"?

Fixed.

12. Line 196: "schematic diagram".

Fixed.

13. Line 221 and throughout: despite widespread use, there is no such thing as "primary sequence" - there is no secondary sequence is there? Please do not be among the 100s of researchers who lack attention to detail and mix up "primary structure" and "amino-acid sequence" into a hybrid term that makes no sense. Just use "amino-acid" or "protein" sequence instead, is there anything wrong with those?

Our apologies, it's fixed.

14. Line 259-278: what sort of predictions does one get when using AlphaFold Multimer to predict an oligomer of at least 6 monomers? This would be another way to analyze the differences between different PYDs, as long as the control predictions reproduce the experimental structures.

Thanks for the suggestion. As shown in the figure below, we tried this with 6, 12, 18, and 24 IFI16^{PYD} protomers. Only one solution from using 24 protomers provided a filamentous model seemingly similar to our experimentally determined structure (circled). However, upon aligning it to our structure, we found that the overall helical architecture still deviates quite a bit. Thus, it appears that, despite the tremendous advancement, employing AlphaFold Multimer for studying PYD•PYD interactions is not quite there yet.

6 protomers

12 protomers

18 protomers

24 protomers

cyan: experimentally determined IFI16^{PYD} filament
dark_pink: AlphaFold prediction (circled above)

15. Line 385-390: is there any good evidence that IFI16 triggers ASC filament assembly at all (i.e. the third possibility would be it triggers the inflammasome using a different set of proteins)? I guess the first possibility with a "bridging" protein also involves an indirect link (although "bridging" paints a picture of a direct bridge). Please rewrite this discussion to allow all possibilities and refer to evidence why some may be more likely than others.

We rewrote this part. (lines: 321-327).

16. Line 468: I suggest "assays".

Fixed.

17. Several references have incomplete information: all 18. Supp Fig 1: the diagram makes no sense, the two parallel paths meet only at the structure - surely they don't lead to the same structure independently. Also make capitalization consistent.

Fixed.

19. Supp Fig 2: what is "Lorem ipsum"?

Our apologies. Deleted.

Reviewer 2

A. The authors were apparently very excited about this finding, reiterated some of the messages multiple times. Perhaps they would consider move line 93-101 in front of 82, and merge discussion line 340-360 together with the introduction, so that the key message could be emphasized directly in the discussion, since IFI16 PYD is concluded not relevant to ASC activation based on the provided data.

We reworked the introduction and rewrote these sections (lines: 101-104).

B. Divergent and convergent, as well as distinct surface pairing/seeding were seen in many CARD domain mediated signaling events, thus perhaps the statements in line 76-78 could be tuned down a bit to better reflect the current DD research landscape. I will recommend incorporating citations (Nature Communications volume 12, Article number: 189 (2021), figure 6; and Nature Communications volume 12, Article number: 188 (2021), figure 4). The authors also mentioned that the new interfaces might indicate the presence of novel interaction partners, perhaps they could shed light on such possibilities. (STING signaling pathway?)

Our apologies for not properly indicating these important studies. We cited them and reworked these sections (lines: 94-98). We also speculated the function of IFI16 in the cGAS-STING pathway in Discussion (lines: 328-333).

C. Since most of the supporting evidence were lack of interaction with ACS inflammasome PYDs, negative data could also be explained by misfolding and non-ideal recombinant constructs. Filament formation of small globular domains sometimes could be completely driven by concentration or certain buffer conditions, like many enzymes and viral proteins. The authors are at the best positions to comment on potential oligomeric filamentous structures of IFIx and

MNDP PYD domains, any preliminary biochemical information could significantly strengthen the conclusion. Of course, complete dissection the biological significance of the observed new filament architecture is a highly demanding job, could be outside this manuscript's scope.

Thanks for the suggestion. We tested whether IFI16^{PYD} and MND1^{PYD} would co-localize with ASC^{PYD} when con-transfected into HEK293T cells. As seen from IFI16, IFI16^{PYD} filaments did not co-localize with ASC^{PYD}. MND1^{PYD} did not form filaments and failed to co-localize with ASC^{PYD} filaments. These findings are included in Figure EV2D and mentioned in lines: 242-246.

Reviewer 3:

1. The introduction needs to more comprehensively cover IFI16 structure and biology (paragraph on lines 82-104). The introduction would benefit from descriptions of IFI16 PYD features and properties. Citations should be included for more of the studies/groups that made initial discoveries of IFI16 "assembling filaments on unchromatinized dsDNA" (lines 82-84) and "accentuating cGAS-STING-dependent interferon signaling pathways, viral replication restriction, and even chromatin dynamics regulation during DNA damage responses" (lines 87-88). For example, more structural studies should be referenced on IFI16 binding to DNA through its HIN200 domains (ex. Jin et al., 2012, PMID: 22483801; Ni, et al., 2016, PMID: 26246511). cGAS-dependent stabilization of IFI16 was shown to facilitate innate immune responses during herpes simplex virus infection (Orzalli et al., 2015, PNAS). IFI16 was shown to be an antiviral innate immune factor and viral restriction factor, both earlier and in response to a much wider range of viruses than the included Almine, et al., 2017, Hotter, et al., 2019 and Jonsson, et al., 2017 papers (ex. Li, et al., 2012, PMID: 22691496; Orzalli, et al., 2012, PMID: 23027953 ; Li, et al., 2013, PMID: 24237704 ; Thompson, et al., 2014, PMID: 33986530 ; Cigno, et al., 2015, PMID: 25972554 ; Jiang, et al., 2021, PMID: 33986530). Primary literature on IFI16 association with DNA damage responses should be included (ex. Dunphy, et al., 2018, PMID: 3019309 ; Justice, et al., 2021, PMID: 34144993).

Our apologies for neglecting these important studies. We added them and reworked the introduction (lines: 111-114).

2. In their previous study (Garg et al., 2023), they show that IFI16FL is unable to promote the polymerization of ASCPYD regardless of the bound nucleic acids. In this study, is there a specific reason why they choose a condition where IFI16PYD, but not IFI16FL, auto-assembles into filaments without any additional molecules (e.g., nucleic acids) for the subsequent structural analysis?

The IFI16^{FL}•dsDNA complex has been practically impossible to work with for structural studies, as these filaments are very prone to aggregation (e.g., PMID: 30232276; we have been trying this for last 10+ years). Please also see below.

3. Another concern is whether this represents a relevant biological context. Regardless of whether IFI16 functions to activate the inflammasome, in most cases, there are stimuli or binding partners that IFI16 interacts with to induce the downstream signaling. Aside from characterizing the IFI16PYD self-assembly filaments, do the authors think the IFI16PYD filaments would be similar in a cellular context with full-length IFI16 and binding molecules? Does IFI16 typically form distinct filaments in cells without any binding partners? In Figure 5, the authors propose a model

illustrating how the atypical structure of IFI16 filaments may participate in non-inflammasome functions, including controlling gene expression, regulating chromatin dynamics, and restricting viral replication. These functions all require an interacting partner (primarily nucleic acids) with IFI16FL.

Thanks for the insightful comment. To our knowledge, virtually all existing high-resolution PYD (and even CARD) filament structures are obtained by using isolated domains. Importantly, we and others have found that these PYD filament structures are very much physiologically relevant, as conclusions/predictions from studying such PYD filaments have been consistent with the experimental results using full-length proteins. For example, as shown in Figure 4, AIM2^{PYD} and AIM2^{FL} co-localize with ASC^{PYD} and ASC^{FL}, respectively; however, both IFI16^{PYD} and IFI16^{FL} fail to do so. (please see also: e.g., PMIDs: 33980849, 33420028, 24630722, 30674671, 35559676, 36442502). Additionally, prior studies have consistently demonstrated that ligand binding at sensor domains such as HIN200 and DEAD-box helicase domains provides a scaffold for PYD/CARD filament assembly (increasing the probability of oligomerization by placing multiple sensors in proximity; e.g., PMID: 30449722, 38374299, 25018021, 26859869).

4. Figure 4D, 4E: The authors transfect different forms of ASC and IFI16 constructs into HEK293T cells and observe no colocalization. However, the localization state of the transgenes may be altered by their relative overexpression levels. It would be helpful to repeat the same experiment in cells with endogenous levels of IFI16 and ASC localized within the same cellular compartment. Otherwise, it would be helpful to perform titration experiments that could conceivably cover an expected range of the endogenous expression levels.

Thanks for the suggestion. As Reviewer knows (and for its namesake), IFI16 is subject to overexpression by IFN and its endogenous concentration can vary several folds (e.g. PMID:20480502, 22046441, 30232276). Per Reviewer's suggestion, we used multiple plasmid amounts in our transfection studies, all of which did not show co-localization with ASC. This new data is shown in Figure EV2C/E and mentioned in lines: 241-242; 253-254

Minor points:

1. Based on their structural analysis of IFI16PYD filaments and existing literature on IFI16 post-translational modification, the authors should briefly comment on which modifications could potentially disrupt or reshape the helical architecture of the filaments, and which sites might be the most significant.

We added them in Discussion (pg15, lines: 322-334)

2. I believe the paper cited in line 285 (Li et al, 2013, Cell Host Microbe) should be a different paper, Li, et al., 2012, PNAS (PMID: 22691496).

Our apologies. It's fixed.

3. Based on point mutations made in Figure 3, the authors determined that IFI16PYD filament assembly relies on its primary sequence. This is consistent with previous literature examining IFI16 PYD-dependent filamentation upon generating PYD point mutations (ex. PMID: 31337724 ; 37283074), which should be referenced. While the authors mutated side chains at the filament interfaces conserved in IFI16-related ALRs but not in AIM2, surrounding regions of the

primary sequences are also not always similar between IFI16 and AIM2. It would be interesting to discuss or test whether replacement of partial/entire helices would restore IFI16PYD filaments (ex. expression of an IFI16 mutant with dual-replaced K12 and D71 helical regions).

We cited the two papers (lines: 160-161). Also, thanks for the suggestions. Indeed, compared to single mutants, we found that the K12T/D71K double mutant partially preserved the filament forming activity of IFI16^{PYD}. This new data is shown in Figure EV2A and mentioned in the text lines: 212-217.

Dear Dr. Sohn,

Thank you again for the submission of your revised manuscript (EMBOJ-2025-120548R) to The EMBO Journal for our consideration, and for your patience during peer review. I apologize for the rather slow re-review process in this case, which was due to the limited availability of the referees during the summer holiday season. Thank you for your patience and understanding.

Your revised manuscript has been sent back to the three original referees who had previously assessed the first version of your manuscript, and we have now received their comments, which you can find below. I am very pleased to say that, as you will see, all three referees are satisfied with the revision. They explain that the study is novel, significant for the field, and robust, and they all recommend publication in The EMBO Journal. In light of this input, I am glad to inform you that your manuscript has been accepted for publication here in principle.

There are only a few minor suggestions for further improvement of the text by referee #2 (regarding re-organization of the literature citations in the Introduction and Discussion; the addition of a schematic to facilitate the Discussion; the balanced acknowledgement of the study's limitations; and an addition to the Discussion on how the missing domains may participate in the overall biological activities of IFI16/IFIx/MNDA), which I would like to ask you to address in a final version of your manuscript. When you are ready, please submit the final version to our manuscript submission system along with a point-by-point response to the referees' comments and detailing all changes to the manuscript.

There are also a few changes and corrections from the editorial side we kindly request you to address in this final version of your manuscript, before we can move forward with its formal acceptance and publication:

- The funding information provided in the Acknowledgements section of the manuscript should be identical to that entered in our manuscript submission system; currently, the following information is missing from the online system: "Johns Hopkins Accelerator Award".
- Please provide a list of up to 5 keywords (preferably broad terms to enhance the online search engine discoverability of your article) after the Abstract of your revised manuscript.
- For all deposited datasets in external repositories that are mentioned in your Data Availability statement, please make sure that they will be publicly available at the time of publication, and provide for each one the database, the accession number-ID, and the specific and permanent URL.
- Please change heading "Conflict of Interest" to "Disclosure and competing interests statement".
- The author contributions statement should be removed from the manuscript file. Instead, we use CRediT to specify the contributions of each author in the journal submission system. Please feel free to use the free text box to provide more detailed descriptions during submission. See also our guide to authors for more information: <https://www.embopress.org/page/journal/14602075/authorguide#authorshipguidelines>.
- We noticed that callout(s) for Figure 2C are missing.
- Please note that no response has been selected from the drop-down menu in section "Experimental study design and statistics" - question about blinding, in your Author Checklist.
- The legend for the uploaded Dataset should be provided in a separate tab/sheet of the same Excel file.
- Heading "Materials and Methods" should be changed to "Methods".
- The Reagents and Tools table should be removed from the main manuscript file, and only be uploaded as a separate file (as it already is).
- Please consider adding some annotation/labels to the synopsis image, to make it more informative for our broad readership. In addition, please remember that this image must be exactly 550 pixels wide, while its height is variable (in the 300-600 pixel range). Please make sure that all text in the image will be legible at the final dimensions.
- Please note that information related to "n" is missing in the legend of Figure 3E.
- The order of manuscript sections must be corrected as follows: Title page - Abstract and Keywords - Introduction - Results - Discussion - Methods - Data Availability - Acknowledgements - Disclosure and Competing Interests Statement - References - Figure Legends - main Tables (Table 1 in this case) - Expanded View Figure Legends.

Please also note that as part of the EMBO publications' Transparent Editorial Process, The EMBO Journal publishes online a Peer Review File along with each accepted manuscript. This File will be published in conjunction with your paper and will include the referee reports, your point-by-point response and all pertinent correspondence relating to the manuscript. You can opt out of this by letting the editorial office know (contact@embojournal.org). If you do opt out, the Peer Review File link will point to the following statement: "No Peer Review File is available with this article, as the authors have chosen not to make the review process public in this case."

We look forward to seeing a final version of your manuscript as soon as possible. Please let us know if you have any questions and use this link to submit your revision: <https://emboj.msubmit.net/cgi-bin/main.plex>.

Best regards,

Ioannis

Referee #1:

The authors have adequately addressed the reviewers' comments in the revised manuscript.

Referee #2:

The authors have demonstrated sincerity in addressing reviewers' comments, particularly the point-to-point response. (This courtesy could be better reflected in the updated main text) The introduction and the summary of the research background has been updated by incorporating the recommended references. Experimental wise, more results were included to enhance the novelty of the alternative complex formation of IFI16/IF1x/MNDA's PYD domain. The key structural message (recombinant IFI16 PYD forms a different filament, other than canonical ASC-PYD filament) is adequately strengthened and properly defended. For a structural story it is sufficiently robust.

I have two suggestions to be considered by the authors:

1. Readers of EMBO journal with alternative research background, like cell biology and genetics, may dislike the treatment of references in the introduction and discussion. And this may hinder the general audience's interest in taking this manuscript seriously and relate the key message to their own research topics. One example, when addressing reviewer's comments, the authors added a lot of references all together (line 111-114), without separating these diverse and sometimes conflicting biological observations apart. Similar cases in discussion when speculating alternative roles of IFI16/IF1x/MNDA, more specific quotation of literature could be done. Investing this additional effort explaining and elaborating in more detail would allow other researchers to appreciate the findings better.
2. Biological context and full length IFI16. Schematic domain map of IFI16/IF1x/MNDA could be included in either Figure 1 or Figure 3 (alongside with sequence alignment) to facilitate the discussion. It is understood that due to practical reasons, the authors only studied PYD filament. The limitations shall be properly acknowledged. All structural investigations were performed in a highly artificial, purified, reconstructed environment anyway. Discussing how the missing domains may participate in the overall biological activities of IFI16/IF1x/MNDA would be interesting to many other researchers.

Referee #3:

The authors have thoughtfully addressed all prior concerns and comments. This manuscript makes an important contribution in the field and should be well received by the scientific community.

All editorial and formatting issues were resolved by the authors.

Dear Dr. Sohn,

Congratulations on an excellent manuscript! I am very pleased to inform you that it has been accepted for publication in The EMBO Journal. Thank you for comprehensively addressing the initially raised referee criticisms and the editorial requests for corrections and changes.

If you have any questions, please do not hesitate to contact the Editorial Office. Thank you for your contribution to The EMBO Journal. Working with you has been a pleasure.

Best regards,

Ioannis
